

# Potassium-limitation of forest productivity, part 1: A mechanistic model simulating the effects of potassium availability on canopy carbon and water fluxes in tropical eucalypt stands

Ivan Cornut[1,2,5], Nicolas Delpierre[1,3], Jean-Paul Laclau[2,5], Joannès Guillemot[2,4,5], Yann Nouvellon[2,6], Otavio Campoe[6], Jose Luiz Stape[7], Vitoria Fernanda Santos[8], and Guerric le Maire[2,5]

[1]Université Paris-Saclay, CNRS, AgroParisTech, Ecologie Systématique et Evolution, 91405, Orsay, France.
[2]CIRAD, UMR Eco&Sols, F-34398 Montpellier, France
[3]Institut Universitaire de France (IUF)
[4]Department of Forest Sciences ESALQ, University of São Paulo, Piracicaba, São Paulo, Brazil
[5]Eco&Sols, Univ. Montpellier, CIRAD, INRAe, Institut Agro, IRD, Montpellier, France
[6]Universidade Federal de Lavras, Departmento de Ciências Florestais, Lavras, MG, Brazil
[7]Department of Forest Science, São Paulo State University, 18610-034 Botucatu, SP, Brazil
[8]Suzano Papel e Celulose, Brazil

**Correspondence:** Ivan Cornut (ivan.cornut@cirad.fr)

**Abstract.** The extent of the potassium (K) limitation of forest productivity is probably more widespread than previously thought, and K-limitation could influence the response of forests to future global changes. To understand the effects of K-limitation on forest primary production, we have developed the first ecophysiological model simulating the K cycle and its interactions with the carbon (C) and water cycles. We focused on the limitation of the gross primary productivity (GPP) by K availability in tropical eucalypt plantations in Brazil. We used results from large-scale fertilisation experiments as well as C

flux measurements in two tropical eucalypt plantations to parameterize the model. The model was parameterized for fertilised conditions and then used to test for the effects of contrasting additions of K fertiliser. Simulations showed that K-deficiency limits GPP by more than 50% during a 6-year rotation, a value in agreement with the literature. The negative effects of K-deficiency on canopy transpiration and water use efficiency were also reported and discussed. Through a sensitivity analysis,

we used the model to identify the most critical processes to consider when studying K-limitation of GPP. The external inputs of K to the stands, such as the atmospheric deposition and weathering fluxes, and the regulation of the internal fluxes of K within the ecosystem were critical for the response of the system to K deficiency. Litter decomposition processes were of lower importance. The new forest K-cycle model developed in the present study includes multiple K processes interacting with the carbon and water cycles, and strong feedbacks on GPP through forest growth were outlined.

**Keywords**

GPP, Potassium, Eucalypt plantations, K-limitation, process-based modelling, Brazil



# 1 Introduction

Nutrient limitation of plant growth has been well-established since the 19th century (Liebig, 1841). Several macro- (N, K, P) or micro-nutrients can limit the growth of plants (Townsend et al., 2011). The nitrogen (N), phosphorus (P) and potassium

(K) limitation of plant growth is a well established phenomenon, as demonstrated by the widespread use of NPK fertilisers in agriculture. It has however less extensively been studied in natural ecosystems. This probably stems from the fact that, contrary to agrosystems where field trials are currently set up to select the best fertilisation regimes, natural ecosystems, and particularly forests, are rarely fertilised. Counter-examples in forestry include lime application trials (Bonneau, 1972; Guitton et al., 1988; Rocha et al., 2019) and other fertilisation trials (Hyvönen et al., 2008). This limitation of primary production by nutrients will

get more palpable as the atmospheric concentration of $CO_2$, one of the substrates limiting photosynthesis, increases (Terrer et al., 2018; Jonard et al., 2014; Penuelas et al., 2020).

N and P are generally considered to be the most limiting elements for global forest growth. N-limitation is supposedly more widespread in temperate and boreal forest ecosystems, while tropical forest are thought to be more P-limited (Du et al., 2020; Cunha et al., 2022; Manu et al., 2022). This paradigm neglects other macro- and micro-nutrients as causes of limitation or

co-limitation. In the tropics, evidence from *Eucalyptus* plantations in Brazil suggests that K and micro-nutrients are often the primary limiting elements for productivity (Silveira et al., 2000; Cornut et al., 2021). More generally, the K-limitation of forest growth appears to be a widespread phenomenon, which has been overlooked so far (Tripler et al., 2006; Sardans and Peñuelas, 2015). Beyond its role on forest growth, K is also an element of geopolitical importance (Nardelli and Fedorinova, 2021) since it is an essential component of most agricultural fertilisers and potash sources are spread among few countries (Prakash and

Verma, 2016).

Despite its importance for forest ecosystems, few models have so far been developed to investigate the K cycle in forest ecosystems. Some models focused on the impact of anthropogenic perturbations and management on multiple nutrient cycles (Johnson et al., 2000), and among them the cycle of K in temperate forests (e.g. models NuCM, Liu et al. (1992); ForNBM Zhu et al. (2004)). Potassium models for annual crops have also been developed and focused mainly on the K dynamics in soils

and uptake by the plants (Seward et al., 1990; Silberbush and Barber, 1984). To the best of our knowledge, only one K model, developed for arable crops, has to date formalised the link between K availability and plant productivity, through an empirical relationship (Greenwood and Karpinets, 1997). This feedback had previously been deemed necessary to predict K uptake more accurately (Seward et al., 1990). Beside these studies, which explicitly modelled the ecosystem K cycle at a broad scale, some papers have quantified through ecophysiological modelling the sensitivity of ecosystem functioning to the availability of K.

For example, the influence of K on the Gross Primary Productivity (GPP) (Christina et al., 2015) and water fluxes (Christina et al., 2018) of tropical *Eucalyptus* plantations has been quantified with the MAESPA model, using a specific parameter set for each of the K-fertilised / non-fertilised treatment. In these works, the K cycle was not explicitly modelled.

Modelling the various aspects of the ecosystem cycle of K is a worthwhile endeavour, since K influences the ecosystem water and carbon cycles in many ways (Cornut et al., 2021). Indeed, K availability has a strong influence on the canopy

photosynthesis (i.e. the *source* of carbon for the plant) through its role on leaf development and senescence. Under low K





availability, leaf expansion is reduced by up to 30% (Battie-Laclau et al., 2013) and leaf lifespan is strongly reduced, with estimated reductions from 25% (Laclau et al., 2009) up to 50% (Battie-Laclau et al., 2013). The resulting loss in leaf area, combined to K-deficiency anthocyanic (purple) symptoms that diminish the leaves' photosynthetic capacity (Battie-Laclau et al., 2014a), lead to a strong reduction of GPP (Epron et al., 2012). While it is more challenging to study the activity of the

plant's carbon *sinks* (i.e. the transport and use of carbohydrate molecules for the maintenance of tissues, growth, constitution of reserves and defense (Körner, 2015)), there is evidence that assimilates transport processes are also influenced by the availability of K. For example, the loading and unloading of sugars from the phloem are affected by K deficiency (Marschner et al., 1996), and more generally, the K nutritional status of the tree has an impact on phloem sap mobility (Epron et al., 2016). Anthocyanic symptoms that develop on leaf margins could in particular be the consequence of the lower ability of K-deficient

leaves to export sugars into the phloem sap (Landi et al., 2015). This body of evidence points towards a strong sink-limitation (mostly through the alteration of phloem export capacity) of GPP under K limitation in addition to a source limitation due to a reduced leaf area. More details relating to the influence of K on these sink and source processes can be found in Cornut et al. (2021). On the topic of the water cycle, it has been shown that K concentration in the xylem sap has an effect on the xylem conductivity (through a change of xylem pit conductivity, Nardini et al., 2010). Potassium deficiency also impacts stomatal

functioning (Marschner, 2011) but an absence of effect of K deficiency on intrinsic WUE has been shown in tropical eucalypt stands (Epron et al., 2012; Battie-Laclau et al., 2016).

The combined influences of K on C-source and C-sink processes explain the K limitation of productivity. The present study focuses on modelling the influence of K on the C-source (i.e. on GPP), which is a prerequisite before modelling the influence of K on C-sink processes, and a first step to start modelling the K limitation on productivity. Process-based models of the C-source

activity have been developed for more than four decades (Farquhar et al., 1980), which contrasts with models representing the activity of C-sinks (e.g. (Hölttä et al., 2006)) which, while relevant (e.g. Guillemot et al., 2017; Körner, 2015), are relatively new and have not been validated at a large scale. While the N- and P-limitation of GPP have been considered in models at scales from the leaf to the globe (Thum et al., 2019; Goll et al., 2012, 2017; Yang et al., 2014), no process-based model simulating the K cycle and its influence on GPP has been published so far.

The objectives of the present study were thus to:

1. develop a model of the K biogeochemical cycle, coupled to the carbon and water cycles, in forest ecosystems,

2. evaluate the model using carbon and water flux data measured at an eddy-covariance site installed in a fertilised (+K) tropical Eucalypt plantation,

3. quantify the influence of K availability on the carbon (gross primary productivity, GPP) and water (evapotranspiration) 
ecosystem-atmosphere fluxes and on the water-use efficiency of a tropical Eucalypt stand, through simulations in non-limited +K stands and stands with omission of K fertiliser (oK),

4. conduct a sensitivity analysis of the model, with the aim to identify the main processes responsible for the response of GPP to the availability of K at the stand level.





To this end, we have developed a new K circulation module in an existing ecophysiological forest model and represented
the response of different physiological processes to the availability of K in the plant. The model was parameterized and tested
on tropical *Eucalyptus* plantations. Because those ecosystems have a continuous phenology, it required the creation of a leaf
cohort model (see e.g. (Sainte-Marie et al., 2014)) that explicitly takes into account the effect of K on different leaf level
processes (leaf expansion, lifespan, etc.).

## 2 Materials and Methods

### 2.1 Study Sites

#### 2.1.1 Eddy-covariance site (EUCFLUX)

The EUCFLUX site is located within a 200 ha plantation located in south-eastern Brazil (São Paulo State, 22°58'04" S and
48°43'40"W, 750 m asl), and is managed under a cooperative project of the IPEF (Instituto de Pesquisas e Estudos Florestais)
(Nouvellon et al., 2019). The precipitation was on average 1536 mm year$^{-1}$ (from 2008 to 2017), with a drier season between
June and September, and the mean annual temperature was 19.3°C. Soils are deep Ferralsols (>15 m). A clonal plantation of
a fast growing *Eucalyptus grandis* × *urophylla* hybrid was established in November 2009 and harvested in June 2017. At the
centre of the stand, a flux tower continually measured meteorological variables as well as the fluxes of $CO_2$ and water vapour
between the plantation and the atmosphere, with the eddy covariance method (Baldocchi, 2003). The study area was described
in details in Christina et al. (2017); Nouvellon et al. (2010, 2019); Vezy et al. (2018). The stand was fertilised in November
2019 with 3.0 g/m$^2$ of $K_2O$, 3.3 g/m$^2$ of $P_2O_5$, 1.8 g/m$^2$ of N, 400 g/m$^2$ of dolomitic lime, and trace elements, then at 3 months
with 3.6 g/m$^2$ of $K_2O$, 3.12 g/m$^2$ of N, at 10 months with 6.72 g/m$^2$ of $K_2O$, 3.08 g/m$^2$ of N and at 20 months of age with
15.12 g/m$^2$ $K_2O$. This amounted to a total of 23.60 gK.m$^{-2}$ from fertilisation and resulted in non-limiting nutrient availability
for tree growth. This value was higher than the typical 12 gK.m$^{-2}$ added on average in commercial plantations (Cornut et al.,
2021).

#### 2.1.2 Fertilisation experiments (Itatinga)

A 2 ha split-plot fertilisation trial was installed at the Itatinga experimental station (23°02'49"S and 48°38'17"W, 860 m asl,
University of São Paulo-ESALQ). It is located 12km aside of the EUCFLUX site, under similar climate and soil conditions.
A fast growing *Eucalyptus grandis* clone was planted in June 2010 and the soil/tree relationships were studied over the entire
rotation of 6 years (from planting to harvesting). The experimental design was described in detail in Battie-Laclau et al. (2014b).
Six treatments (three fertilisation regimes and two water supply regimes) were applied in three blocks. In the present study,
we focus on the +K and oK treatments with the undisturbed rainfall regime, which consisted in a non-limiting fertilisation +K
(17.55 gK.m$^{-2}$ applied as KCl at planting, with 3.3 gP m-2, 200 g m-2 of dolomitic lime and, trace elements, as well as 12 gN
m-2 at 3 months of age) and an omission treatment oK where the same fertilisers were applied as in +K treatment, except K.
The area of each individual plot in the experiment was 864m$^2$.



The concentrations of different elements (N, P, K) in the organs (leaves, trunks, branches, and roots) were measured at an annual time step in 8 individuals of each fertilisation treatment and upscaled to the whole stand using allometric relationships (not shown). Biomass and nutrient contents were calculated (using upscaling) from inventories, biomass and nutrient concentration measurements conducted at 1, 2, 3, 4, 5 and 6 years in each fertilisation treatment. Atmospheric deposition (0.55 $gK.m^{-2}.yr^{-1}$) and canopy leaching fluxes (0.42 $gK.m^{-2}.yr^{-1}$) were measured in a nearby experiment from Laclau et al. (2010).

## 2.2   Complementary foliar measurements


Area, mass and K-deficiency symptom development of individual leaves were measured for the studied sites to parameterize the new leaf cohort sub-model and the K-deficiency-symptom area sub-model described below. To this aim, we used the scan pictures (tabletop scanner device model HP Scanjet G4050, 300 dpi) of leaves collected during the biomass samplings at both sites (every six-months at EUCFLUX and annually at Itatinga), on at least 6 trees per date and treatment and at three

crown levels. Individual leaf areas as well as the proportion of anthocyanic symptoms on individual leaves were automatically computed from the images. The leaf-scale metrics were up-scaled to stand averages using linear regressions with individual tree $D^2H$ (i.e. the product of squared diameter with tree height), for each canopy thirds. Regression were done using the *scikit-learn* python library (Pedregosa et al., 2011). The resulting parameters and functions were then applied to the $D^2H$ of trees using inventories of diameter and height of plots. This allowed the upscaling of leaf individual area and symptomatic leaf area

in order to compute their plot averages.

## 2.3   CASTANEA-MAESPA general model presentation

The soil-vegetation-atmosphere carbon and water balance were simulated with the CASTANEA-MAESPA model for the EUCFLUX and Itatinga Eucalypt plantations. CASTANEA-MAESPA was the merging of the CASTANEA model (Dufrêne et al., 2005) with the MAESPA model (Duursma and Medlyn, 2012), the latter being modified as in Christina et al. (2017). CAS-

TANEA is an ecophysiological model simulating the fluxes of carbon and water between a forest stand (average tree) and the atmosphere at an half-hourly time step. In its basic version, it includes no representation of the hydraulic soil-plant-atmosphere continuum, which is however critical in the context of a coupled carbon-water-potassium model. The MAESPA model (Duursma and Medlyn, 2012) was developed using the above-ground components of the MAESTRA model (Wang and Jarvis, 1990) and the water balance components of the SPA model (Williams et al., 1996). MAESPA is a three dimensional model of

light interception, energy balance, photosynthesis and evapotranspiration. These fluxes are computed from prescribed description of individual trees along time, and at the scale of small volumes of leaves within each tree crown. The soil-plant-atmosphere water continuum is explicitly simulated by MAESPA.

    It was not possible to adapt the CASTANEA model, initially developed on temperate Beech (*Fagus sylvatica*) forests (Dufrêne et al., 2005), to the particular study case of tropical *Eucalyptus* plantations, as we did previously for several temperate

and Mediterranean species (e.g. Delpierre et al., 2012; Davi et al., 2006; Le Maire et al., 2005). Indeed, tropical *Eucalyptus* plantations can grow roots down to a depth of 6m the first year after planting (Christina et al., 2011), which violates the CASTANEA assumption of a constant rooting depth over the simulation period, and the use of a simple soil water bucket model.





The MAESPA model does not have this constraint and can easily be adapted to simulate an increasing amount of extractible water (Christina et al., 2017). Moreover, MAESPA had already been parameterized and applied at the EUCFLUX and Itatinga

sites (Christina et al., 2015, 2017). However, although it simulates fluxes of carbon and water, MAESPA, is not a full carbon balance model, in the sense that it does not simulate the carbon allocation within the plant, litterfall, soil organic matter decomposition, etc. As such, contrary to CASTANEA, MAESPA does not provide alone the structure required to simulate the K balance. Therefore, the merging of both models in CASTANEA-MAESPA model aimed at offering a relevant and extensive ecophysiological model for C and water cycles in eucalypt plantations, prior to the implementation of the K processes as

described below.

The modules of CASTANEA simulating light interception, water interception, carbon allocation and the growth of organs and organ respiration were coupled with the modules of MAESPA simulating soil water dynamics, leaf photosynthesis, transpiration, and plant hydraulics. Note that in the coupled model, the leaf photosynthesis module of MAESPA was applied to canopy layers of CASTANEA. As such, the coupled model takes the 1-D vertical structure of CASTANEA. This simplified as-

sumption of canopy horizontal homogeneity is valid in these even-aged clonal plantations at least after canopy closure Binkley et al. (2013).

## 2.4 Model of *Eucalyptus* canopy dynamics

### 2.4.1 Overview of the leaf cohort model

Highly productive tropical eucalypt plantations in Brazil grow from seedlings to 25-30 meter high trees in the span of 6-7

years. The plantations present a continuous foliar phenology with leaf production and leaf fall throughout the year. This has previously led to the development of a canopy dynamics model (Sainte-Marie et al., 2014). While this model was sufficient to explain leaf production and leaf fall dynamics, we found it necessary to develop a new cohort-based canopy dynamics model (Summarised in Fig.1). The creation of this model stemmed from the need for the simulation of both K cycling in the canopy and the effects of K on foliar ontogeny. We chose to implement leaf cohorts as the elementary objects for the simulation of

the canopy dynamics. The use of a leaf cohort model stemmed from the continuous phenology of *Eucalyptus* trees as well as the importance of K on leaf ontogeny (Laclau et al., 2009; Battie-Laclau et al., 2013). A daily time step was necessary for the simulation of expansion and fall of the leaves of each cohort. All leaves within a cohort were considered to have the same physiological characteristics, growth and lifespan. A cohort was characterised by a number of leaves per square meters of ground, individual leaf area and mass. This new leaf cohort model is described in the next sections, in the case of no limitation

by K.

### 2.4.2 Leaf cohort production

A new cohort was initialised daily. The number of leaves $N$ produced in the cohort was a function of the height increase of the trees. Indeed, in these fast-growing plantations, most of the new leaves are produced in the top-most part of the crown. The



increase in tree height can be computed in the CASTANEA-MAESPA model as the result of increase in trunk biomass, and
with allometric parameters relating stand biomass and stand height (see the companion paper: Part 2).

The relationship between daily height increase and leaf production was corrected by a flattening factor. This means that even
if the daily height increase was close to zero or even null, leaf production would still happen at a slower but positive rate. The
model generated a number of new leaves per m$^2$ at a daily time-step following this function:

$$N = \frac{\Delta H + f_p}{1 + f_p} \times \kappa \tag{1}$$

where $N$ was the daily number of leaves produced in number of leaves per m$^2$ and $\Delta H$ (m) was the increase in tree height. $f_p$
was the flattening factor, meaning that if $f_p = 0$ then leaf production was linearly related to height increase and as $f_p$ increased,
$P_{leaf}$ tended towards a constant function. $\kappa$ is a conversion factor from height increment to number of new leaves in number
of leaves per m of vertical growth per m$^2$ of ground. The parameters used here were fitted using experimental data from the
fully fertilised stand.

### 2.4.3 Leaf cohort lifespan

As long as K was not limiting, the lifespan of a cohort was considered to be constant since the leaf lifespan deduced from
leaf biomass and leaf fall measurements in fully fertilised stand did not show major trends along the rotation and amplitude
of seasonal changes in lifespan was limited (Fig.S3e). Since no mechanistic explanation was available, we refrained from
implementing it in the model. For the sake of simplicity, we did not consider in the present simulations the fall of leaves
resulting from extreme events (drought, frost, heatwave). Indeed, in the studied sites no large leaf fall due to extreme events
were observed. Here, the leaf lifespan (LLS) in non-limiting K conditions was fixed to the average measured value of 480 days.
This value was estimated by calculating the leaf biomass turnover using yearly biomass and weekly litter measurements.

### 2.4.4 Leaf expansion in area in the cohort

For a given cohort, individual leaf area *LA* expands from a virtually null area at initialisation of the cohort, up to an area of
$LA_{max}$ (mm$^2$). The leaf area followed a sigmoid function (Fig.2a, Battie-Laclau et al., 2013). Leaf area was a function of
time and not thermal time (as for instance in the original CASTANEA model) since no calibration data were available and it
was not deemed necessary for this model. Therefore, the daily leaf area expansion was forced to follow the sigmoid derivative
function:

$$\frac{\Delta S}{\Delta t} = \frac{k_{LA} \times LA_{max} \times e^{-k_{LA}(t-t50_{LA})}}{(e^{-k_{LA}(t-t50_{LA})} + 1)^2} \tag{2}$$

where $\frac{\Delta LA}{\Delta t}$ (mm$^2$.day$^{-1}$) was the daily growth in area of an individual leaf within a given cohort, $t$ (days) was the number of
days since leaf cohort creation, $LA_{max}$ (mm$^2$) was the (non-limited) maximum leaf area, $k_{LA}$ (days$^{-1}$) was a slope parameter,
$t50_{LA}$ (days) was the inflexion point of the original sigmoid of leaf area increase, therefore was the date of maximum leaf
area increase, and also the date when half $LA_{max}$ was reached. The parameters $LA_{max}$, $k_{LA}$ and $t50_{LA}$ were fitted from





measured leaves in expansion (Battie-Laclau et al., 2013) in non-limited fertilisation conditions. Parameters $k_{LA}$ and $t50_{LA}$
were assumed not to vary along the stand rotation. $LA_{max}$ was also assumed to be constant since the leaf scans did not show
any explainable trends of mean leaf area during the rotation (Fig.S4).

The total leaf area of a given cohort was given by the product of $S$, the area of an individual leaf and $N$, the number of leaves
in the cohort. The total leaf area of the stand at a given date was calculated by adding up all the cohort areas.

### 2.4.5   Leaf expansion in mass in the cohort

Individual leaf mass increase within a cohort was similar in shape to the leaf area increase, but with a temporal shift since leaf
mass per area continues to increase when the maximum leaf area is attained:

$$\frac{\Delta BF}{\Delta t} = \frac{k_{BF} \times BF_{max} \times e^{-k_{BF}(t-t50_{BF})}}{(e^{-k_{BF}(t-t50_{BF})} + 1)^2} \tag{3}$$

where $\frac{\Delta BF}{\Delta t}$ (g.day$^{-1}$) was the daily growth in mass of an individual leaf in a given cohort, $t$ (days) was the number of days
since leaf cohort creation, $BF_{max}$ (g) was the maximum individual leaf mass, $k_{BF}$ (day$^{-1}$) was a slope parameter, and $t50_{BF}$
(days) was the inflexion point of the original sigmoid of leaf mass increase, therefore it was the date of maximum leaf area
increase, and also the date when half $BF_{max}$ was reached. The parameters $k_{BF}$ and $t50_{BF}$ were calibrated using individual
leaf biomass data and results from Laclau et al. (2009).

Specific leaf area (SLA) of individual leaves showed a decreasing relationship with tree height (Fig.S5a), while $LA_{max}$ was
more constant as described before (Fig.S4). We thus assumed that $BF_{max}$ increased with tree height:

$$BF_{max} = min\left(BF_{max}^{rotation}, s_{BF} \times H^P\right) \times TC \tag{4}$$

where $BF_{max}$ (gC) was the maximal mass of an individual leaf of a cohort at the end of leaf expansion in mass, $BF_{max}^{rotation}$
(gDM) was the maximum mass of an individual leaf throughout the rotation, $s_{BF}$ and $P$ were the parameters of the power
function between leaf mass and tree height H (m), and TC (gC.gDM$^{-1}$) was the leaf carbon content.

### 2.4.6   Leaf water content

In non-limited nutrient conditions, leaf cell expansion in area was associated with a leaf water inflow in order to maintain an
optimum leaf turgor. This water inflow was computed as:

$$W_{xylem \rightarrow leaf} = \Gamma \times \frac{\Delta S}{\Delta t} \tag{5}$$

where $W_{xylem \rightarrow leaf}$ (mL.day$^{-1}$) was the water inflow into the expanding leaf (this was "structural" water associated to the
creation of new tissues, not to be confounded with the water used for leaf transpiration), $S$ the leaf area of the cohort (mm$^2$),
computed in eq. 2, and $\Gamma$ (mL.mm$^{-2}$) was the surfacic water content, i.e. the amount of leaf water per leaf area at full turgor.
$\Gamma$ was assumed to be a constant.





Experimental data have shown that at the end of leaf area expansion, when the leaf has reached its maximum area, there was some water outflow, defined hereafter as water expulsion. This is an assumption made from observations of a slight decrease in K leaf content following the end of leaf expansion (Laclau et al., 2009). This leaf water (containing ions) expulsion, probably

corresponding to a loss of cell wall extensibility (Pantin et al., 2012) during the maturation of leaf tissue, was limited in quantity and in duration. Hence the overall leaf water content dynamic starts increasing until a maximum at the end of the leaf area expansion, followed by a small decrease until a constant plateau. This plateau corresponds to the water content necessary to maintain a constant leaf turgor in optimal conditions. The water expulsion flux was computed as:

$$W_{leaf \to phloem} = -min\Big(\alpha \times (1 - \frac{W_{leaf}}{W_{leaf}^{turgor}}), 0\Big) \tag{6}$$

where $W_{leaf \to phloem}$ (mL.day$^{-1}$) was the flux of water leaving the leaf at the end of leaf expansion, $\alpha$ (mL.day$^{-1}$) the rate of water expulsion, $W_{leaf}$ (mL) was the amount of water in an individual leaf in previous day, and $W_{leaf}^{turgor}$ (mL) the amount of water found in the leaf at the final plateau. $W_{leaf}^{turgor}$ was computed as $\Gamma \times LA_{max}$.

Finally, the variation of leaf water content for an individual leaf in a cohort ($W_{leaf}$, in mL) was computed by adding the daily net flow $\frac{\Delta W_{leaf}}{\Delta t}$ given by:

$$\frac{\Delta W_{leaf}}{\Delta t} = W_{xylem \to leaf} - W_{leaf \to phloem} \tag{7}$$

## 2.5 Ecosystem model of the K cycle

We now introduce the *CASTANEA-MAESPA-K* model, which simulates K cycling in the plantation, and its interactions with the ecosystem carbon and water cycles (Fig.1). K remains in its ionic (K$^+$) form throughout the cycle (Marschner, 2011). A model of K circulation within the plant, as well as between the plant and the soil, was necessary since K$^+$ is a cation of high

mobility (Marschner, 2011). Similarly to the leaf cohort model, a daily time step was used for the K cycle model. The K cycle was modelled using seven explicit K pools (Fig.1): soil K (subdivided in the fractions of soil K available and not available for root uptake), soil K fertiliser added (the fertiliser before dissolution), litter K, xylem sap K, phloem sap K, leaf K and other plant organs K. These K pools were connected by fluxes (root uptake, resorption, leaching, etc.), and K inputs (fertilisation, atmospheric deposition and rock weathering) entered this open system (Fig.1).

K entered into the soil through fertiliser inputs, atmospheric deposition and rock weathering. After uptake by roots, K circulated throughout the plant trhough the xylem and the phloem, which provided the K necessary to the leaves and organs as well as the K needed for phloem functioning. Part of the K in the phloem was recirculated back into the xylem and thus created a feedback for aK uptake by roots. Indeed, soil K uptake by roots depends on the gradient between soil and xylem K. Leaves contribute to the cycle through resorption, canopy leaching and litterfall. The K in the litter was leached following a rate that

depended on throughfall precipitation amount. It then entered the soil, to be once again available for uptake. The only outgoing flux from the system is the amount of K lost by deep leaching, and the trunk K exported from the stand at harvest. K was accumulated in organs (trunk, branches, roots) but this allocation sub-model will be presented in the companion paper (Part 2).





This K cycle allowed us to create a feedback between K availability and GPP through the effect it has on leaf expansion, leaf lifespan and photosynthetic parameters (see below).

**Figure 1.** Schematic representation of the soil and plant components of the K cycle, and their links with the leaf cohort model and other sub-models. Purple boxes are K state variables, and purple arrows are K fluxes. K fluxes simulated with a simple Ohm's law form are represented with resistance symbols. Black arrows represent a functional link a variable has with another variable or process. The numbers in exponent correspond to the numbers of the equation in the text.

### 270    2.5.1    Soil K

Soil K content ($K_{soil}$, in gK.m$^{-2}$) was initialised in the model at the tree planting date (EUCFLUX: 07/10/2009, Itatinga: 01/06/2010) with a measured value $K_{soil}^{t_0}$, calculated using K concentration in soil, and soil bulk density at different depths (Maquère, 2008). Then, this value was updated daily with incoming and outgoing fluxes.



The K that is added daily to the K litter pool ($K_{litter}$) is the K reaching to the ground through leaf fall, bark fall, branch fall
and entering the soil litter pool through fine root turnover (see Part 2). Instead of a fixed decomposition rate of K in litter, the
model considered K release from litter to be mainly coming from leaching with water since K is an cation, that is not strongly
adsorbed on organic surfaces. Litter K release measurements done at the experimental site (Maquère, 2008) showed very close
K release rates for branches, bark and leaves, further confirming this hypothesis. Moreover, K is released faster than either C,
N or P contained in the litter, suggesting a leaching process independent of litter decomposition. The following equation was
used for K leaching from the litter to the soil:

$$K_{litter \rightarrow soil} = \sigma \times P_{ground} \times K_{litter} \tag{8}$$

where, $K_{litter \rightarrow soil}$ (gK.m$^{-2}$.day$^{-1}$) was the litter K leaching flux, $P_{ground}$ (mm.day$^{-1}$) was the daily amount of precipitation
reaching the ground, $\sigma$ (mm$^{-1}$) was the conversion factor between the K litter leaching rate and throughfall precipitation, and
$K_{litter}$ (gK.m$^{-2}$) was the amount of K in the litter. $\sigma$ was estimated on annual data by dividing the measured K leaching rate
(Maquère, 2008) by the annual precipitation that falls on the ground (throughfall).

K fertilisation was applied at the beginning of the rotation at several dates, in a solid form (crystals of KCl), and located
close to the Eucalyptus plants. The flux of K from this solid fertiliser compartment ($K_{fertiliser}$ in gK.m$^{-2}$) to the soil K
compartment was simulated using the following equation:

$$K_{fertiliser \rightarrow soil} = s_{fertiliser} \times K_{fertiliser} \tag{9}$$

where $K_{fertiliser \rightarrow soil}$ (gK.m$^{-2}$.day$^{-1}$) was the the flux of K from the fertiliser compartment to the accessible soil K pool, and
$s_{fertiliser}$ the decomposition rate of K fertiliser in day$^{-1}$. Observations in the fields showed that the KCl fertiliser dissolved
quickly at EUCFLUX and Itatinga (less than two months).

Atmospheric K deposition is modelled as a constant flux. We used the values measured at Itatinga (Laclau et al., 2010). They
amounted to a mean input of $K_{atmosphere \rightarrow soil}$ of 0.55 gK.m$^{-2}$/year distributed uniformly throughout the year. This amount
feeds directly into the total $K_{soil}$ pool.

Deep soil K leaching was included in the model, but was parameterized to be a null flux, as was measured in the plantations
under study (Maquère, 2008). The K entrance to the soil pool from mineral weathering was simulated as a constant flux. K flux
from weathering is directly added to the accessible soil since this process mainly takes place in the rhizosphere (Pradier et al.,
2017). However, as for deep leaching, there is no clear evidence of this flux in the soils under study, where values between 0
and 0.3 gK.m$^{-2}$/year are given (Cornut et al., 2021) : we therefore also set this flux to zero.

K foliar leaching ($K_{leaves \rightarrow soil}$) was computed within the Leaf K submodel, described below (eq. 27).

Only a portion of $K_{soil}$ was accessible to the roots at the beginning of the rotation because of the time spent for root
horizontal and vertical expansion. Because K was mainly located in the top soil layers (Maquère, 2008), and because root
growth in depth was very fast (Christina et al., 2011), only the horizontal root exploration was considered in the model. An
empirical relationship between tree height and area root radius around individual trees was described in Gonçalves (2000):

$$Root_{Radius} = 0.80 \times H - 0.075 \tag{10}$$



where $Root_{Radius}$ (m) was the average radius of the horizontal root front around a tree and $H$ (m) the tree height (Part 2). Since the planting density was 1666 trees/ha, a full exploration of the soil was obtained when tree had explored a circle of 6 m$^2$-area:

$$K_{soil}^{accessible} = \frac{(Root_{Radius})^2 \times \pi}{6} \times K_{soil} \qquad (11)$$

where $K_{soil}^{accessible}$ (gK.m$^{-2}$) was the soil K accessible for plant uptake, $K_{soil}$ (gK.m$^{-2}$) was the total bioavailable soil K. The fraction is the ratio of root accessible soil to total soil, bounded between 0 and 1.

Because of the root exploration dynamics, the initial K in the system $K_{soil}^{t_0}$ was progressively available to roots, at a proportion following the increase in the root explored area. Following the same logic, the amount of K coming from the litter decomposition and atmospheric deposition entered the total soil K pool $K_{soil}$, but only a part of this $K_{soil}$ was available for plant uptake (called $K_{soil}^{accessible}$). However, the three other incoming fluxes of K to the soil were considered to be directly accessible for root uptake, i.e. they enter directly in the $K_{soil}^{accessible}$ pool: 1) the fertiliser flux since fertilisers are applied close to trees at planting or when the root system is exploring the whole volume of the upper soil layers for other fertiliser applications; 2) the K flow coming from soil weathering because most of the weathering takes place in the rhizosphere (Pradier et al., 2017; de Oliveira et al., 2021); and 3) the canopy leaching flux because it enters the soil mostly below the crown foliage.

### 2.5.2 Uptake of soil K and cycling in xylem and phloem

To calculate the K uptake by trees in the soil and the fluxes of K in the plant it was necessary to calculate the optimal quantity of K in the phloem sap. Furthermore, K in phloem sap is essential to a wide range of processes (e.g. loading/unloading of sugars)(Cornut et al., 2021). For these processes, the plant maintains a fairly constant K phloem sap concentration $[K]_{phloem}$. To compute this K quantity in the phloem, values of optimal K concentration in the phloem sap ($[K]_{phloem}^{opti}$), minimum K concentration in the phloem sap ($[K]_{phloem}^{min}$) and phloem sap volume ($V_{phloem}$) per unit surface were needed.

$[K]_{phloem}^{opti}$ was considered to be the maximum concentration of K in the phloem sap measured in the fully fertilised stand (Battie-Laclau et al., 2014b). $[K]_{phloem}^{min}$ was assumed to be the minimum concentration of K in the phloem sap measured in the K omission stand of the same experiment (Battie-Laclau et al., 2014b).

Estimating $V_{phloem}$ was done through relationships between phloem sap volume and xylem sap volume since no direct measurements or estimates were available. Xylem sap volume was considered to be a function of basal area, sapwood area at DBH (Guillemot et al., 2021), height of the tree, and branch and root biomasses. The trunk cross section was divided in sapwood area and heartwood area. The trunk (respectively heartwood) volume was modelled as a cone with a base disk of area equal to the basal area (respectively equal to the heartwood area). Trunk sapwood volume was estimated as the difference between trunk volume and heartwood volume. Branch and root sapwood volume were deduced from their biomass, considering that branches and root biomass are entirely composed of sapwood. Their volume were computed using the density of *Eucalyptus* sapwood. The lumen volume of the xylem (i.e. the xylem sap volume) was considered to be 13.6% of total xylem volume as reported in general for Angiosperms (Zanne et al., 2010) since no *Eucalyptus*-specific data were available. Following Hölttä



et al. (2013), and considering the relatively similar lumen proportion between both xylem and phloem (Nobel, 2005), phloem
sap volume was considered to be 2% of the total xylem sap volume.

Uptake of K from the soil by the trees was a function of demand by growing organs, remobilisation of K from senescent organs, and soil supply. The amount of K available for uptake was computed in eq. 11. K demand by the trees needs to be calculated. To that end:

First, the target amount of K in the phloem was calculated as:

$$K^{tar}_{phloem} = [K]^{opti}_{phloem} \times V_{phloem} + K_{NPP} + K^{demand}_{leaf} \tag{12}$$

where $K^{tar}_{phloem}$ was the target amount of K in the phloem sap in $gK.m^{-2}$, $[K]^{opti}_{phloem}$ was the optimal K concentration in the phloem sap in $gK.L^{-1}$, $V_{phloem}$ was the volume of phloem sap in $dm^3.m^{-2}$, $K_{NPP}$ was the optimal quantity of K needed for organ growth and $K^{demand}_{leaf}$ was the optimal quantity of K needed for leaf development.

Finally the demand for K uptake from then soil is the following:

$$K^{demand}_{soil \to xylem} = \frac{K^{tar}_{phloem} + K^{tar}_{xylem} - \left( K_{phloem} + K_{remob} + K_{xylem} \right)}{\Delta t} \tag{13}$$

where $K^{demand}_{soil \to xylem}$ ($gK.m^{-2}.day^{-1}$) was the quantity of K uptake necessary for optimal tree functioning, $K^{tar}_{phloem}$ from eq.12, $K^{tar}_{xylem}$ ($gK.m^{-2}$) was the target amount of K in the xylem sap, $Kphloem$ ($gK.m^{-2}$) was the amount of K in the phloem sap, $K_{remob}$ ($gK.m^{-2}$) was the amount of K remobilised from the woody organs (see part 2) and $K_{xylem}$ ($gK.m^{-2}$) was the amount of K in the xylem.

Uptake of K from the soil to the xylem sap is the minimum between the soil "offer", i.e. what can be uptake from the soil knowing the soil K content and the soil to root K resistance, and the xylem K "demand":

$$K_{soil \to xylem} = min\Big( \frac{K^{accessible}_{soil}}{R_{soil \to xylem}}, K^{demand}_{soil \to xylem} \Big) \tag{14}$$

where $K_{soil \to xylem}$ ($gK.m^{-2}.day^{-1}$) was the uptake flux, $K_{soil}$ ($gK.m^{-2}$)the amount of K in the accessible soil, $R_{soil \to xylem}$ (days) the resistance to absorption by plant roots, and $K^{demand}_{soil \to xylem}$ ($gK.m^{-2}.day^{-1}$) the uptake demand from eq. 13.

In the model, internal K cycling (Marschner et al., 1996) was a necessary process that provides feedback for the uptake of K from the soil, maintaining K homeostasis in the phloem sap and linking organ remobilisation and allocation of K for growth. In the K circulation model (Fig.1), two K fluxes are represented, one from the phloem sap to the xylem sap (representing a flux mainly happening in roots *in planta*) and one from xylem sap to phloem sap (mainly happening in the shoots). These representations allowed the phloem sap to maintain a K content of phloem close to optimal values (eq. 12).

Firstly, the flux of K from the xylem sap to the phloem sap was calculated. It was a function of phloem "demand" and xylem sap K of the previous time step. We assumed that all the K available in the xylem sap could potentially be transferred to the phloem sap the next day:

$$K_{xylem \to phloem} = min\Big( max(\frac{K^{tar}_{phloem} - K_{phloem}}{\Delta t}, 0), \frac{K_{xylem}}{\Delta t} \Big) \tag{15}$$




where $K_{xylem \rightarrow phloem}$ (gK.m$^{-2}$.day$^{-1}$) was the flux of K from the xylem to the phloem, $K_{xylem}$ (gK.m$^{-2}$) the amount of K in the xylem sap, $K_{xylem}$ (gK.m$^{-2}$) the amount of K in the phloem sap, and $K_{phloem}^{tar}$ from eq. 12.

The transport of K from the phloem to the xylem took place if K concentration in the phloem sap was higher than its optimal value (e.g. following leaf resorption):

$$K_{phloem \rightarrow xylem} = max\Big(\frac{K_{phloem} - K_{phloem}^{tar}}{\Delta t}, 0\Big) \tag{16}$$

where $K_{phloem \rightarrow xylem}$ (gK.m$^{-2}$.day$^{-1}$) was the flux of K from the phloem to the xylem, and $K_{phloem}^{tar}$ was from eq. 12.

### 2.5.3 K cycling in the leaves

The leaf K balance equation of the individual leaf of each leaf cohort was given by the following sum of fluxes:

$$\frac{\Delta K_{leaf}}{\Delta t} = K_{phloem \rightarrow leaf} - K_{leaf \rightarrow soil} - K_{leaf \rightarrow phloem} - K_{leaf \rightarrow litter} \tag{17}$$

where $\frac{\Delta K_{leaf}}{\Delta t}$ (gK.day$^{-1}$) was the daily variation of the quantity of K in an individual leaf of a given cohort, $K_{phloem \rightarrow leaf}$ was the amount of K entering the leaf during leaf expansion (see eq. 22), $K_{leaf \rightarrow soil}$ was the canopy leaching flux (see eq. 27), $K_{leaf \rightarrow phloem}$ was the sum of K following water expulsion at the end of leaf expansion (eq. 24), the maximum between K resorption driven by the phloem demand (eq. 25) and the K resorption at leaf senescence (eq. 26), and $K_{leaf \rightarrow litter}$ was the K flux occurring the last day of the cohort, when the leaf was simulated to fall.

Leaf K inflow ($K_{phloem \rightarrow leaf}$) was computed as a function of the K offer by the phloem and K demand for leaf growth at the canopy scale and organ growth at the tree scale.

The calculation of the water inflow in the leaf during leaf expansion was calculated first in the case of no K limitation ($W_{xylem \rightarrow leaf}$ in eq. 5). This allowed the calculation of a theoretical optimal K flux entering the expanding leaf, $K_{phloem \rightarrow leaf}^{nonlimited}$, computed considering an optimal concentration of K in the water entering the leaf, $[K]_{leaf}^{max}$ (gK.mL$^{-1}$). This value was approximated as the maximum concentration found in the leaf water on different measurement campaigns (Battie-Laclau et al., 2013; Laclau et al., 2009). The resulting K flux was the non-limited rate of K entrance in the expanding leaf:

$$K_{phloem \rightarrow leaf}^{nonlimited} = [K]_{leaf}^{max} \times W_{xylem \rightarrow leaf} \tag{18}$$

where $K_{phloem \rightarrow leaf}^{nonlimited}$ (gK.day$^{-1}$) was the maximum entrance of K$^+$ ions in the expanding leaf.

However, restriction of this flux occurs due to the phloem limitation of K supply at canopy scale that may not attain the K demand for optimal growth. A reduction of the K inflow in the leaf was therefore applied if the leaf demand at canopy scale $K_{leaf}^{demand}$ was high compared to the $K_{phloem}$ available (the "offer").

$K_{leaf}^{demand}$ ($gK.m^{-2}$) was the K demand of all expanding leaves of the stand, and was computed as the sum of $K_{phloem \rightarrow leaf}^{nonlimited} \times N$ for all leaf cohorts (with N the number of leaves of each cohort, see eq. 1):

$$K_{leaf}^{demand} = \sum_{i=1}^{t} \Big(K_{phloem \rightarrow leaf\ i}^{nonlimited} \times N_i\Big) \tag{19}$$



To calculate the phloem sap "offer" the following relationship was used:

$$K_{phloem \rightarrow organs} = min\Big(K_{phloem} - [K]_{phloem}^{min} \times V_{phloem}, \ K_{NPP} + K_{leaf}^{demand}\Big) \tag{20}$$

where $K_{phloem \rightarrow organs}$ (gK.m$^{-2}$) was the amount of K available for leaf expansion and organ growth in the phloem sap, $K_{phloem}$ (gK.m$^{-2}$) was the total amount of K in the phloem sap, $[K]_{phloem}^{min}$ (gK.L$^{-1}$) was the minimal concentration of K in the phloem sap, $V_{phloem}$ (L) was the phloem sap volume and $K_{NPP}$ (gK.m$^{-2}$) was the optimal amount of K for organ growth (Part 2), and $K_{leaf}^{Demand}$ (gK.m$^{-2}$) was the demand for optimal leaf expansion.

Then the limitation of K for leaf expansion was calculated as a ratio of available ("offer") K to K demand:

$$L_K = \frac{K_{phloem}^{available}}{K_{NPP} + K_{leaf}^{Demand}} \tag{21}$$

where $L_K$ was the ratio of available K in the phloem sap to demand of K from organ growth and leaf expansion, $K_{phloem \rightarrow organs}$ (gK.m$^{-2}$) was available phloem K (eq. 20), $K_{NPP}$ and $K_{leaf}^{Demand}$ were organ and growth demands (both gK.m$^{-2}$, see above).

The quantity of K entering the expanding leaf was thus defined as the following:

$$K_{phloem \rightarrow leaf} = K_{phloem \rightarrow leaf}^{nonlimited} \times L_K \tag{22}$$

where $K_{phloem \rightarrow leaf}$ (gK.day$^{-1}$) was the amount of K$^+$ ions that enter the expanding leaf in limited K conditions, $K_{phloem \rightarrow leaf}^{nonlimited}$ was computed in eq. 18 and $L_K$ was computed in eq. 21

The K outgoing flux from leaf to phloem (Fig.2b) can be decomposed into:

$$K_{leaves \rightarrow phloem} = K_{expulsion} + max\big(K_{resorption}^{phloem}, \ K_{resorption}^{senescence}\big) \tag{23}$$

where $K_{expulsion}$ (gK.m$^{-2}$.day$^{-1}$) was the K flux leaving the leaf during leaf maturation (eq. 24), $K_{resorption}^{phloem}$ (gK.m$^{-2}$.day$^{-1}$)
was the resorption flux driven by phloem sap demand (eq. 25), and $K_{resorption}^{senescence}$ (gK.m$^{-2}$.day$^{-1}$) was the resorption flux driven by leaf senescence (eq. 26).

$$K_{expulsion} = W_{leaf \rightarrow phloem} \times \frac{K_{leaf}}{W_{leaf}} \tag{24}$$

where $W_{leaf \rightarrow phloem}$ was calculated in eq. 6, $W_{leaf}$ was the previous day leaf water content, calculated in eq. 7, and $K_{leaf}$ was the previous day leaf K content of the cohort.

The K resorption flux $K_{resorption}$, from the leaf to the phloem could be activated by low phloem K content. This was a mechanism to maintain homeostasis in the phloem since K was essential for many phloem functions (Cornut et al., 2021). Evidence was also provided by leaves losing K during their lifespan, especially in K-deficient trees (Battie-Laclau et al., 2013). Another piece of evidence was the high concentrations of K in the petiole compared to other leaf parts (Fig.S8d). This was not the case for N (Fig.S8c) and suggests an intense circulation of K to and from the leaf. The resorption of the leaf towards the





phloem was:

$$K_{resorption}^{phloem} = \frac{K_{leaf}}{R_{leaf \to phloem}} \times (1 - L_K) \tag{25}$$

where $K_{resorption}^{phloem}$ (gK.day$^{-1}$) was the cohort phloem driven resorption, $K_{leaf}$ (gK) was the K content of leaves in the cohort, $R_{leaf \to phloem}$ (days) was the resistance to resorption, and $L_K$ was the K limitation computed in eq. 21.

The leaf K resorption flux during leaf senescence $K_{resorption}^{senescence}$ was fast (Battie-Laclau et al., 2013). The amount of resorbed

K follows a sigmoid function:

$$K_{resorption}^{senescence} = \frac{e^{-k_r(t-LLS)}}{(e^{-k_r(t-LLS)} + 1)^2} \tag{26}$$

where $K_{resorption}^{senescence}$ (gK.day$^{-1}$) was the resorption flux occurring at leaf senescence, just before leaf fall. $LLS$ (days) was the leaf lifespan, which was also the inflexion point of the sigmoid, and $k_r$ was the parameter corresponding to the speed of the resorption flux at the inflexion point. We approximated the time it took for active K resorption to be one week as K$^+$ ions are

highly mobile and evidence from chlorophyll degradation at senescence suggest extremely fast dynamics (Mattila et al., 2018).

We assumed that the daily canopy leaching flux strength was proportional to the throughfall that occurs during precipitation as observed previously in a *Eucalyptus* forest (Crockford et al., 1996):

$$K_{leaves \to soil} = \lambda \times W_{tip} \times K_{leaf} \tag{27}$$

where $\lambda$ ($mm_{throughfall}^{-1}$) was the fraction of leaf K that was leached per mm of daily throughfall, $W_{tip}$ (mm) was the through-

fall and $K_{leaf}$ (gK) was the amount of K in the leaf. $\lambda$ was calibrated considering the leaf area index and leaf K content of a well fertilised canopy as well as canopy K leaching measurements (Laclau et al., 2010).

Finally, the K flux accompanying the leaf fall, $K_{leaf \to litter}$, happened following one of the two conditions: when leaf cohort lifespan $LLS$ was reached, or when the K concentration in leaf water ($\frac{K_{leaf}}{W_{leaf}}$) was below a threshold value $[K]_{min}$ of $9.25e^{-5}$ gK.mL$^{-1}$. At one of these dates the leaf cohort was shed and $K_{leaf \to litter} = K_{leaf}$. This $[K]_{min}$ threshold value was

either reached after resorption during senescence or through other processes (phloem demand, eq. 25; leaching, eq. 27) thus diminishing the leaf lifespan in K deficient trees. Indeed, leaf fall was related to strong K deficiency in several studies (Laclau et al., 2009; Battie-Laclau et al., 2013).

## 2.6    Impact of K limitation on the cohort growth model

### 2.6.1    Number of leaves produced at cohort initialisation

Since leaf production was a function of tree height which itself is a function of tree trunk biomass, K availability could have an indirect impact on leaf production through its impact on tree trunk production (see the companion article Part 2). No specific impact of K deficiency was included in the model since experiments have shown that leaf generation speed at the branch level is not impacted by K availability and leaf biomass production is not substantially different between oK and fully K fertilised stands (Cornut et al., 2021).



### 2.6.2 Impact of K limitation on individual leaf area

When there was no K limitation, in optimal conditions, leaf expansion in area was computed as in eq. 2, and the water inflow was simply simulated to follow this leaf expansion as in eq. 5. However, under K limitation, individual leaf area was strongly affected by K availability (Battie-Laclau et al., 2013). Mechanistically, the increase of leaf area was driven by a water flux entering the leaf, because the turgor pressure participates to the cell expansion, following the logic of the Lockhart model (Lockhart, 1965). The Lockhart model was simplified in the present study due to the important number of parameters of the original model that had not been measured in our context and the difficulty regarding their calibration. This model allowed to relate the K availability in the phloem sap and the expansion of leaves at the individual leaf level on a daily time step. Using the dynamic water content of leaves during expansion, K demand for the cohort at each time step was calculated. The availability of K in the phloem sap then determined a K-limited water flux and thus the leaf expansion rate (Fig.2d).

First, K availability controls the water entrance flux ($W_{xylem \rightarrow leaf}$, eq. 5) in the leaf during leaf expansion since there was a lower limit of osmotic potential required for the entrance of water in the leaf cells.

$$W_{xylem \rightarrow leaf}^{Klimited} = W_{xylem \rightarrow leaf} \times max\left(\frac{K_{phloem \rightarrow leaf}}{K_{phloem \rightarrow leaf}^{nonlimited}}, r\right) \tag{28}$$

where $W_{xylem \rightarrow leaf}^{Klimited}$ (mL.day$^{-1}$) was the flux of water entering the leaf during leaf expansion reduced with K limitation, $K_{phloem \rightarrow leaf}$ from eq. 22 and $K_{phloem \rightarrow leaf}^{nonlimited}$ from eq. 18, and $r$ a parameter $r \in [0,1]$ of the same order of magnitude as the ratio of K limited individual leaf area compared to non-limited leaf area.

Secondly, leaf water content $W_{leaf}$ was re-computed using eq. 7 with $W_{xylem \rightarrow leaf}^{Klimited}$ instead of $W_{xylem \rightarrow leaf}$.

Finally, the non limited leaf area expansion increment $\frac{\Delta S}{\Delta t}$ computed in eq. 2 was updated with a new K limited leaf area expansion increment $\frac{\Delta S_{Klimited}}{\Delta t}$, considered to be directly proportional the water flux entering the leaf:

$$\frac{\Delta S_{Klimited}}{\Delta t} = W_{xylem \rightarrow leaf}^{Klimited} \times \frac{1}{\Gamma} \tag{29}$$

where $\frac{\Delta S_{Klimited}}{\Delta t}$ (mm$^2$.day$^{-1}$) was the area increase of the expanding leaf computed after accounting for K limitation, $W_{xylem \rightarrow leaf}^{Klimited}$ was obtained from eq. 28, and $\Gamma$ ($mL.mm^{-2}$) was the leaf surfacic water content.

## 2.7 Leaf K-deficiency symptoms and implication for leaf photosynthesis

### 2.7.1 Leaf K-deficiency symptoms

When leaves experience strong K deficit, they display anthocyanic symptoms (i.e. they turn purple from the leaf margins, Gonçalves, 2000). This has a strong impact on the photosynthetic capacity of affected areas (Battie-Laclau et al., 2014a). We assumed that leaf K-deficiency symptom area results from the history of K deficiency the leaf has experienced since the beginning of its growth. This was modelled as function of the accumulation of K-deficit in the leaves over time, called "deficit days" ($DD$). The daily increase in $DD$ was computed as:

$$\frac{\Delta DD}{\Delta t} = max\left(([K_{leaf}]_{max} \times W_{leaf}) - K_{leaf}, 0\right) \tag{30}$$



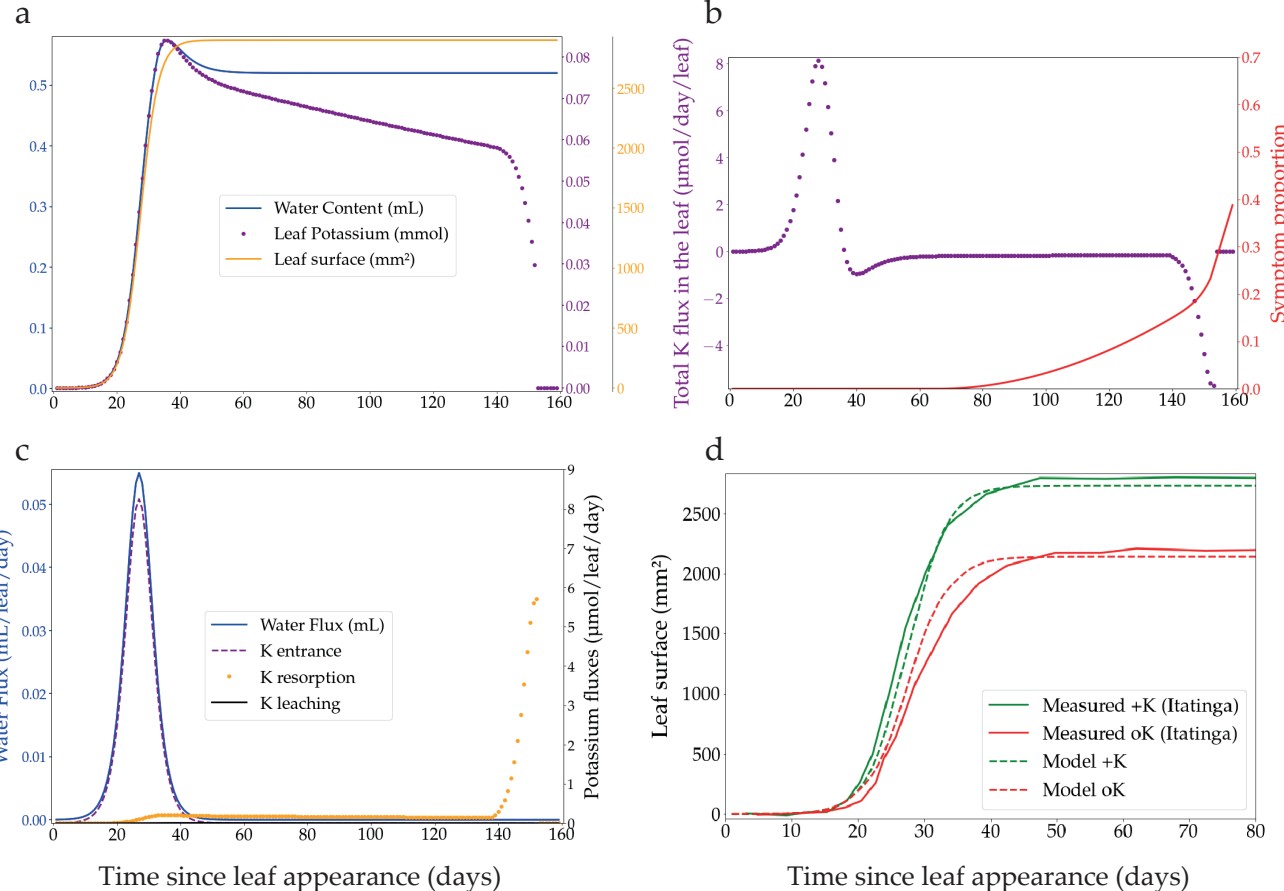

**Figure 2.** Outputs of the theoretical leaf cohort expansion model over the course of the lifespan of a single leaf in the cohort: state variables that are produced by the model (a); fluxes of water and K at the individual leaf scale (b); total flux of K at the leaf scale showing the transition from K sink (positive flux to the leaf) to K source (negative flux from the leaf) (c); comparison of the model to the individual leaf area data from Battie-Laclau et al. (2013) in two contrasted K availabilities (d).

where $\frac{\Delta DD}{\Delta t}$ (g) was the daily increase of the deficit days, $[K_{leaf}]_{max}$ (gK.mL$^{-1}$) was the optimal (maximal) foliar concentration of K, $W_{leaf}$ (mL) was the amount of water in the individual leaf (after K limitation, eq. 7), and $K_{leaf}$ (gK) was the leaf K content.

     The proportion of symptoms in a leaf (Fig.2c) was then computed as:

$$SP = min\Big(DD \times \Theta, \ SP_{max}\Big) \tag{31}$$

where $SP$ was the leaf surfacic symptom proportion, $DD$ was the accumulated deficit days computed in eq. 30, $\Theta$ was a conversion factor from deficit days to symptom proportion and $SP_{max}$ was the maximum proportion of symptom area on a single leaf, with $0 < SP < SP_{max}$.





### 2.7.2 Impact of symptoms on leaf photosynthesis

Leaves, even with symptoms, continue to intercept radiations. In the model, it means that the light interception submodel was
not affected by symptoms area, i.e. the total leaf area of each cohort was not affected by leaf K symptoms. Note that the
total leaf area under K deficiency was reduced through various processes such as lower number of produced leaves because of
lower growth in height (eq. 1), reduction of individual leaf sizes (eq. 29), and through the shorter lifespan of leaves because of
K-deficiency associated leaf fall (section 2.5.3).

However, leaf symptoms have a strong effect on leaf-scale photosynthesis. Indeed, experimental results (Battie-Laclau et al.,
2014a) have demonstrated that the leaf scale photosynthesis was strongly reduced when there was K-deficiency symptoms.
This decrease was almost linear, suggesting that we could model leaf photosynthesis as fully active in the non-symptomatic
areas of the leaves, and null in the symptomatic area, i.e. the photosynthesis was reduced by the proportion of symptoms in the
leaf.

For sake of simplicity, this was implemented in the model by reducing the two leaf scale photosynthetic parameters $Vc_{max}$
and $J_{max}$ according the leaf area proportion affected by symptoms:

$$Vc_{max}^{lim} = Vc_{max} \times (1 - SP) \tag{32}$$

$$J_{max}^{lim} = J_{max} \times (1 - SP) \tag{33}$$

where $Vc_{max}$ and $J_{max}$ were respectively the maximum carboxylation rate and the maximum rate of electron transport, $SP$
was described in eq. 31.

### 2.8 Model parameterisation and initialisation

Most of the parameters of the model were obtained from (Christina et al., 2017), except the parameters of the new cohort
model and of the new K cycle model, for which the parameterisation was described along the equation descriptions of sections
2.4 to 2.7, and reported in Tables S1,2. The beginning of the simulation was considered to be the planting date. Tree height
at planting was set at 10-cm. The canopy was initialised with a very small, but not null, amount of leaves: 10 leaves of 30
mm$^2$ each per m$^2$ (eqv. to 0.0003 m$^2$leaf/m$^2$soil). The soil was divided into 50 layers of either 33-cm (for the 3 top layers of
soil) or 50-cm (the 47 bottom layers) depth each, and soil properties for each layer were obtained from (Christina et al., 2017).
Initial values of water content of the soil on the planting day were set as measured at the EUCFLUX site (Christina et al.,
2017). All model runs were initialised with the amounts of K present in the soil and in the litter compartment. The amounts
of K present in the litter were determined using measurements of the mass and elemental dosages of the litter present on the
ground at the beginning of the rotation in the Itatinga experiment, which amounted to 1.92 gK.m$^{-2}$ (Laclau et al., 2010). The
amount of K present in the soil compartment at the start was deduced from exchangeable soil K concentrations and bulk soil
density measurements from soil surface to a depth of 18m (Maquère, 2008). It amounted to 0.507 gK.m$^{-2}$ (it was converted
from gK.m$_{soil}^{-3}$). The simulations were run on EUCFLUX site.



## 2.9 Sensitivity analysis


A sensitivity analysis was conducted with a One-At-a-Time (OAT) approach, in both K-fertilised (+K) and K-omission (oK) conditions to test the sensitivity of GPP to the different processes. The sensitivity of GPP to all the parameters of the newly introduced sub-models was tested. The method used was the following: in each fertilisation condition (+K and oK) the parameter was increased and decreased by 10%, except the fertilisation parameters which were fixed to their +K and oK treatment values.

The model was then run for each combination of fertilisation and parameter values. The total average GPP of the simulated rotation was compared to the simulated average GPP of the rotation with the same fertilisation regime and the initially fixed value of the parameter. The percentage of difference between +K and oK simulations gave the response of the simulated GPP to the variation of the chosen parameters.

## 3 Results

## 3.1 Ecosystem K cycle


The quantity of K accessible in the soil for the plant was on average 62 times as high in the +K (Fig.3a) compared to the oK fertilisation treatments (Fig.3b). While the K stored in the canopy was only a small fraction (23%) of the total K in the system in the +K stand, leaves accounted for more than half of the total K stock in the oK stand (52%). In both stands, the quantity of K stored in the litter was small, representing 3.8% of total K in +K treatment and 27% in oK (Fig.3, Tab.1). In the +K stand

the amount of K in the leaves increased until 2 years after which it remained steady up to harvesting (Fig.3a). By contrast, the increase only lasted for one year in the oK stand, and was quickly followed by a strong decrease (Fig.3b). The strong decrease in $K_{leaves}$ was concurrent to an important decrease in $K_{soil}^{accessible}$ as the initial litter stock was depleted while the plant demand was still high and a lower leaf biomass in oK. In the +K stand, the fertiliser quickly compensates for the decrease in initial litter K, increasing the $K_{soil}^{accessible}$ to high values.

| Stocks (gK.m$^{-2}$) | +K | oK |
|---|---|---|
| $K_{soil}^{Accessible}$ | 11.18 | 0.16 |
| $K_{litter}$ | 0.59 | 0.20 |
| $K_{canopy}$ | 3.68 | 0.39 |

**Table 1.** Mean value of simulated K stocks over the entire rotation

In the +K treatment, over the course of the rotation, the fluxes of fertiliser, atmospheric deposition, litter leaching (eq. 8) and canopy leaching were respectively 59%, 9%, 28%, 4% of the total amount of K that entered the soil (Tab. 2). In the oK stand, they were respectively 0%, 43%, 56%, 1% (Tab. 2). So while the litter stock was small (Tab. 1), the cumulated flux of K from the litter to the soil was important for K cycling in both fertilisation regimes. In both stands, the resorption flux from leaves ($K_{leaf \rightarrow phloem}$) was more important than the sum of canopy leaching ($K_{leaves \rightarrow soil}$) and litter leaching flux ($K_{litter \rightarrow soil}$,

Tab. 2), highlighting the role of the tree internal K cycling.




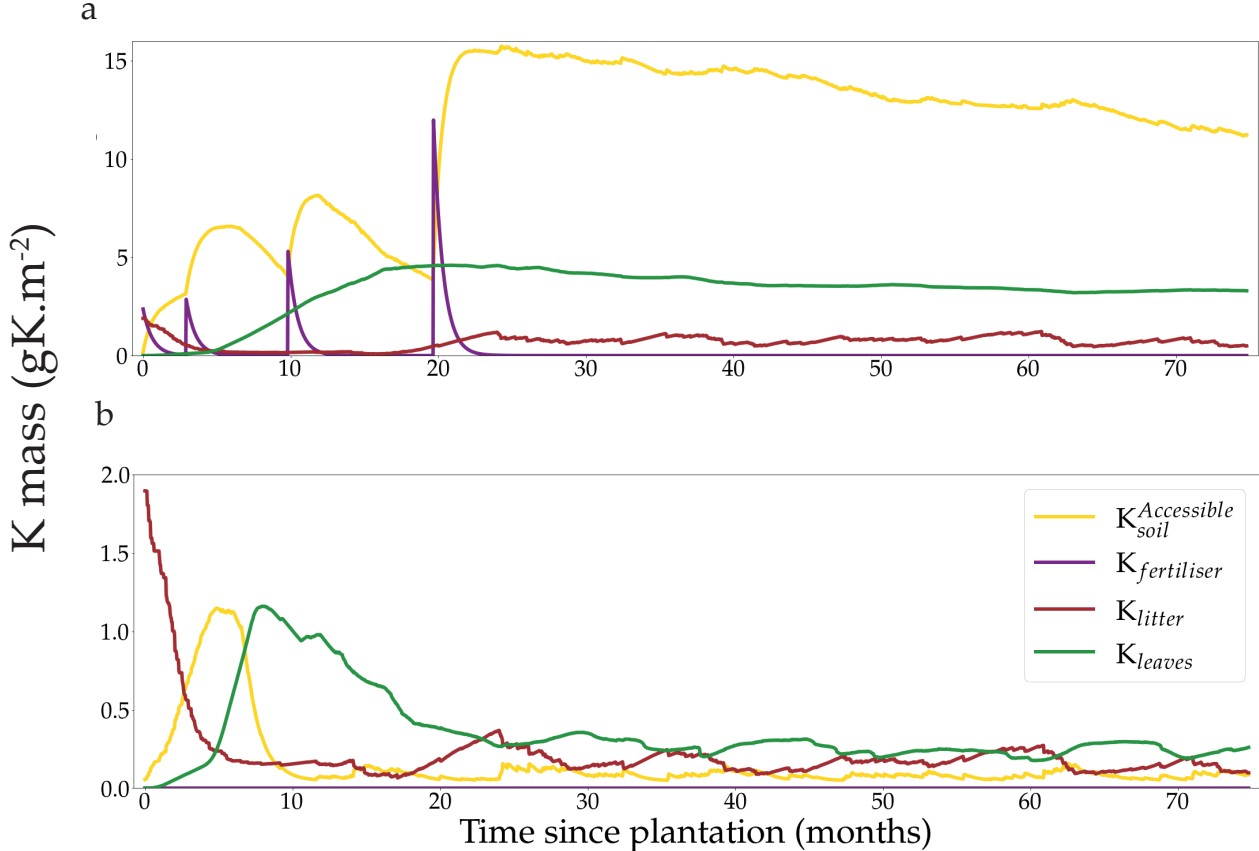

**Figure 3.** Simulated stocks of K in the soil, litter, fertiliser and canopy compartments, in a non K-limited (+K) stand (a) and in the same K-limited (oK) stand (b). Note differences of the y-axis scales for better visualisation.

| Fluxes (gK.m$^{-2}$.yr$^{-1}$) | +K | oK |
|---|---|---|
| $K_{fertiliser \rightarrow soil}$ | 3.60 | 0 |
| $K_{atmosphere \rightarrow soil}$ | 0.55 | 0.55 |
| $K_{litter \rightarrow soil}$ | 1.71 | 0.66 |
| $K_{leaves \rightarrow soil}$ | 0.27 | 0.01 |
| $K_{soil \rightarrow xylem}$ | 4.67 | 1.29 |
| $K_{leaves \rightarrow phloem}$ | 2.04 | 0.77 |

**Table 2.** Mean value of simulated yearly fluxes of K (b) for two contrasted fertilisation regimes: +K and oK.

In the +K treatment, leaf resorption ($K_{leaves \rightarrow phloem}$) was equal to 43% of the average uptake flux ($K_{soil \rightarrow xylem}$, Tab.2). In the oK, this proportion was higher (60%) thus showing the importance of the internal K recycling for the maintenance of a suitable K supply for growing organs.





## 3.2 Leaf cohort model and canopy dynamics

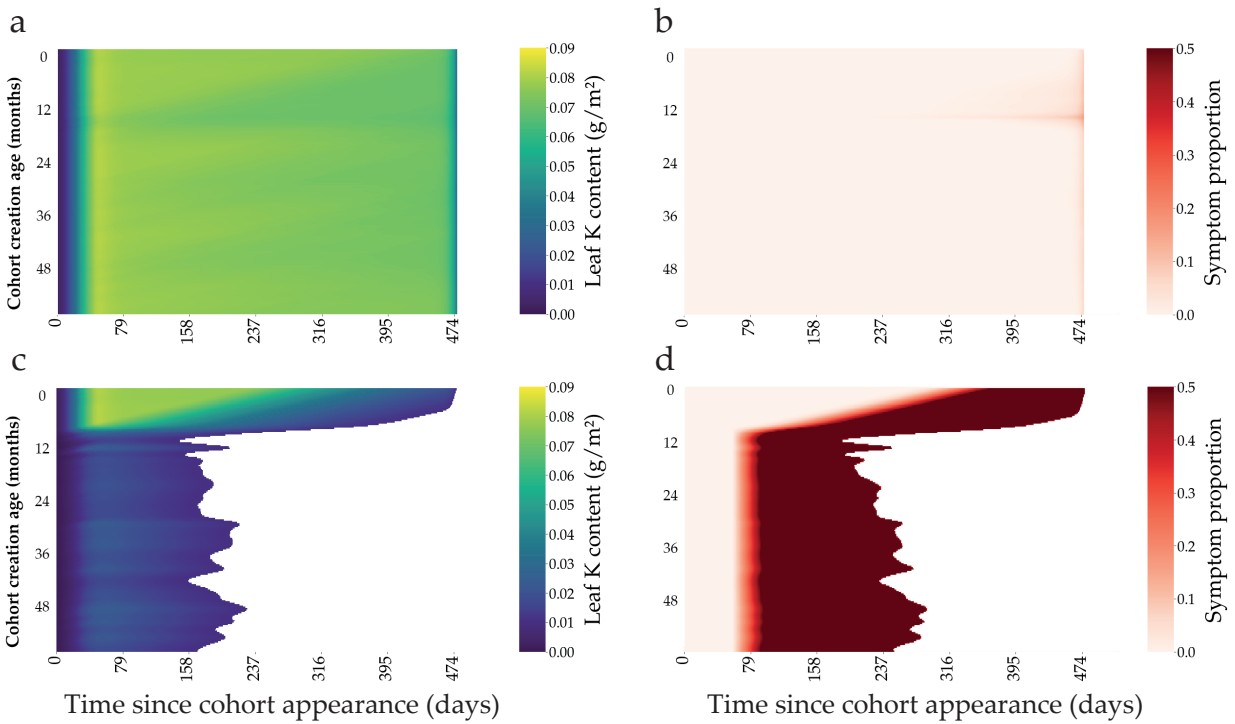

**Figure 4.** Outputs of the leaf cohort model in two contrasted K fertilisation regimes. The K content present in each individual leaf of the cohort is represented through the lifespan of a cohort (x-axis) for the different cohort created along the first 60 months of the rotation (a, c). The symptom area proportion for each leaf of the cohort is also represented (b, d). Top subplots (a, b) were simulated in +K conditions, while bottom subplots were oK simulations (c, d).

The leaf expansion model was successful in simulating the influence of K on both the dynamics and maximum value of the individual leaf area (Fig.2d). Positive fluxes of K into the leaf took place during the expansion process (Fig.2b,c). Foliar leaching, K expulsion after leaf expansion (eq. 6) and resorption were responsible for fluxes of K going out of the leaf across its lifespan (Fig.2b). This model allowed us to represent leaf K content in the leaves at the organ scale and also revealed the variation of K availability at the leaf level during the rotation. In +K condition, K availability was high during the whole

rotation with symptom area proportion of the canopy always below 2.5% (Fig.5b) throughout the leaf lifespan, that reached its maximum ($LLS$, fixed value). On the other hand, in oK simulations, leaf lifespan was greatly reduced (less than half of the leaf lifespan of fertilised stands, Fig.4c) and symptom proportions reached more than 40% during a major part of the rotation (Fig.4d,5b). The patterns of the leaf K content in the different cohorts during the oK rotation had two phases (Fig.4c): a first phase of the rotation where soil K bioavailability was high and a second phase with very low K concentrations in leaves.

This mirrors the K availability in the soil and litter sub-system (Fig.3b). The first phase corresponds to a high initial litter





decomposition flux (litter originating from the preceding rotation which was fertilised with K), in the second phase the only fluxes of K to the soil were the litter leaching flux (recycling) and atmospheric deposition (external input). These cumulated fluxes were not sufficient to satisfy the plant K demand.

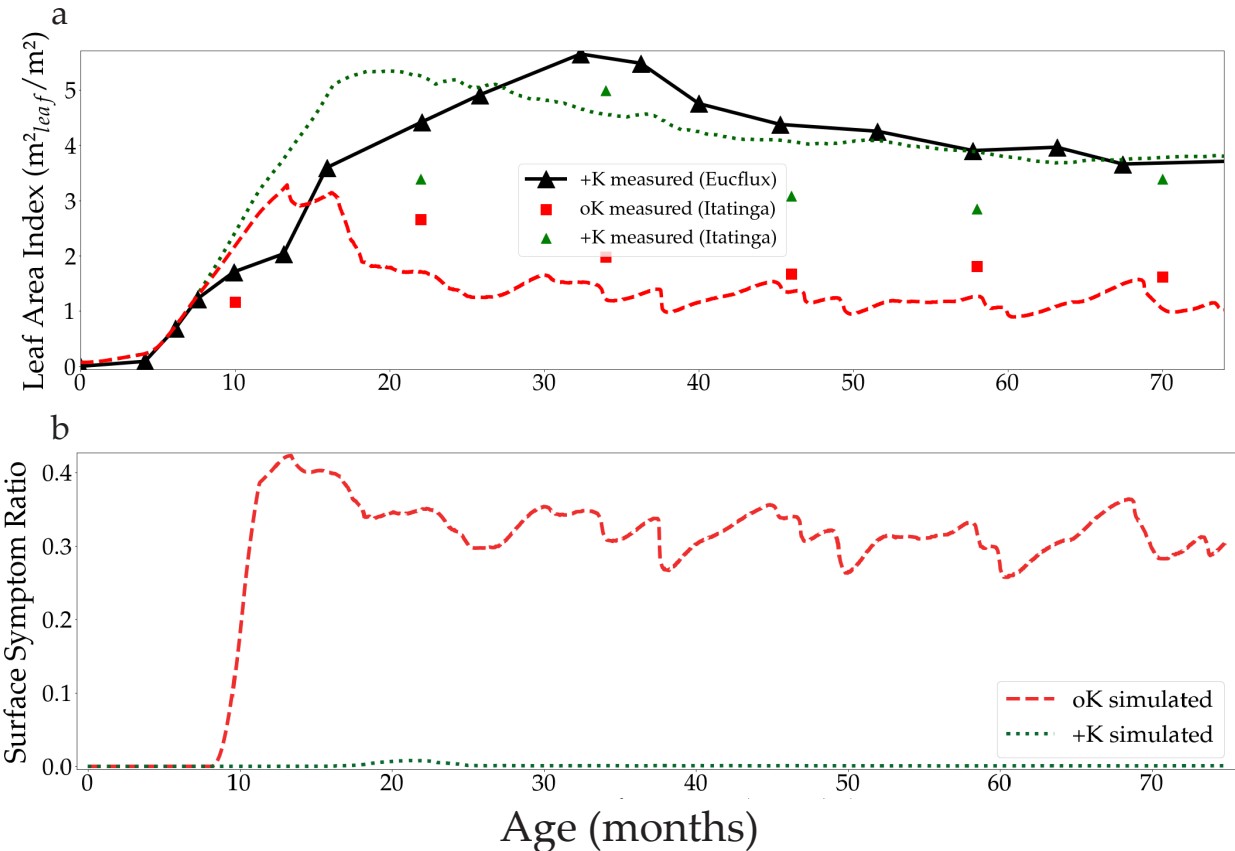

**Figure 5.** Simulated Leaf Area Index in both the fertilised +K and non fertilised oK treatments, and the Leaf Area Index measured at the Itatinga experiment and at the eddy-covariance site EUCFLUX (a). Canopy average proportion of leaf area with symptoms in both fertilisation treatments (b).

The difference of leaf area between the +K and oK simulated stands was higher than observed in the +K and oK treatments
of the Itatinga fertilisation experiments. The mean leaf area of the oK stands were 58% of the leaf area of the +K stand in the experiment versus 43% in the model (Fig.5a). This could be explained by different response to K deficiency between the genetic material (different Eucalypt clones) used at EUCFLUX and at Itatinga. Another possibility was an underestimation of K availability in the oK stand in our simulations. For example, a small change in the mineral weathering flux from 0 to a realistic value of 0.3 $gK.m^{-2}.yr^{-1}$ (Cornut et al., 2021) led to the simulated leaf area in the oK stand being 53% of the +K
stand (not shown).





In the oK condition, symptoms appeared very early during the leaves' lifespan (Fig.4d). The proportion of symptomatic leaf area was slightly higher in the simulations of the EUCFLUX site than measurements at the Itatinga site (Fig.S6). This could be due to an overestimation of the leaves' limitation by K or a difference in response of the genetic material to K availability.

### 3.3 Carbon and water fluxes

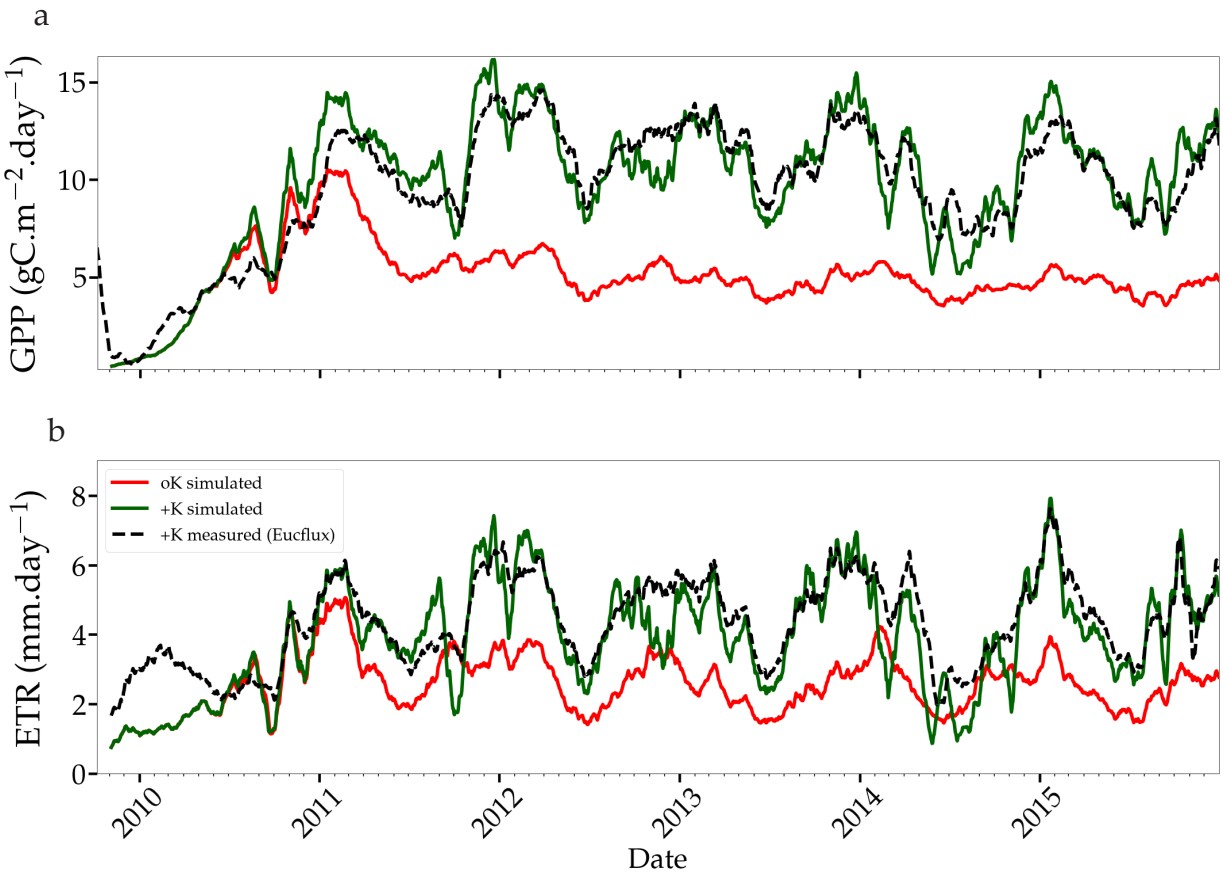

**Figure 6.** Simulated and measured daily gross primary productivity (a) and evapotranspiration (b) fluxes simulated in two stands with contrasted fertilisation regimes. The measurements were performed continuously using the eddy-covariance method at the EUCFLUX site (a +K stand, fully fertilised). A rolling average of 30 days was applied to the observed and simulated time series for sake of clarity.

Simulated GPP was greatly reduced under oK conditions (Fig.6a) and the cumulated GPP in the oK stand was only 50% of the +K stand on average (Table 3). Seasonal fluctuations of GPP between dry and wet seasons were clearly visible in both simulations, however the seasonal variability was higher in the +K stand (Fig.6a) due to lower access to soil water in the +K stand as a result of higher ETR resulting in faster deep soil water depletion (Fig.6b). The difference of GPP between fertilisation regimes was not constant during the rotation. During the first phase (i.e. the first year), the difference was small due to similar





low leaf areas in both fertilisation conditions resulting in low K demand, fulfilled by sufficient K availability for both treatments (Fig.4a,c). The difference was also quite small during the major 2014 drought (Fig.6a) where water-limitation dominated in the +K stand. The simulated GPP were similar to both measurements (Epron et al., 2012) and simulations conducted at Itatinga with the MAESPA model (Christina et al., 2015) (Table 3).

| | Estimated GPP (Itatinga) from Epron et al. (2012) ($gC.m^{-2}.yr^{-1}$) | Modelled GPP (Itatinga) from Christina et al. (2015) ($gC.m^{-2}.yr^{-1}$) | Modelled GPP (EUCFLUX) in this study ($gC.m^{-2}.yr^{-1}$) |
|---|---|---|---|
| Age (years) | +K − oK | +K − oK | +K − oK |
| $0 \rightarrow 1$ | ... − ... | 1300 − 800 (61%) | 1129 − 1083 (95%) |
| $1 \rightarrow 2$ | ... − ... | 3500 − 2500 (71%) | 3926 − 2519 (64%) |
| $2 \rightarrow 3$ | ... − ... | 4600 − 2900 (63%) | 4541 − 1782 (39%) |
| $3 \rightarrow 4$ | ... − ... | ... − ... | 3971 − 1636 (41%) |
| $4 \rightarrow 5$ | 4440 − 2540 (57%) | ... − ... | 3653 − 1670 (45%) |

**Table 3.** Annual GPP at the study sites, under contrasted K supply regimes. Values from Epron et al. (2012) were inferred from fluxes and biomass increment measurements obtained from a previous fertilisation experiment at the Itatinga site. Values from Christina et al. (2015) were simulated by the MAESPA model. Percentages between parentheses indicate the ratio of GPP between the oK and +K treatments for each experiment. The data presented are different from those on Fig.6b that display evapotranspiration.

| | Modelled Transpiration (Itatinga) in Christina et al. (2018) ($mm.yr^{-1}$) | Modelled Transpiration (EUCFLUX) in this study ($mm.yr^{-1}$) | $WUE_{GPP}$ this study ($mmolC.molH_2O^{-1}$) |
|---|---|---|---|
| Age (years) | +K − oK | +K − oK | +K − oK |
| $0.5 \rightarrow 1.5$ | 947 − 654 (69%) | 969 − 858 (88%) | 4.69 - 4.13 |
| $1.5 \rightarrow 2.5$ | 1365 − 881 (64%) | 1605 − 791 (49%) | 4.10 - 3.73 |
| $2.5 \rightarrow 3.5$ | 1438 − 753 (52%) | 1344 − 649 (48%) | 4.38 − 3.95 |
| $3.5 \rightarrow 4.5$ | 1323 − 774 (58%) | 1458 − 678 (46%) | 4.05 − 3.69 |

**Table 4.** Annual transpiration fluxes for contrasting K supply regimes both in our study and in a previous modelling work that used the MAESPA model (Christina et al., 2018). Percentages between parentheses indicate the ratio of transpiration between the oK and +K treatments for each experiment.

Our simulations showed reduced evapotranspiration under K deficiency (Fig.6b), which was expected since K deficiency had

a strong impact on leaf area (Fig.5a). We compared our transpiration simulation results with those obtained using the MAESPA model at the Itatinga oK stand. The MAESPA simulated transpirations had been validated using sap-flow measurements. While in the first part of the rotation the difference between treatments simulated by our model was lower than simulated by MAESPA, in the following years our simulations were close to MAESPA results (Table 4). Total 5-year cumulated transpiration in the oK plot was 54% of that of the +K plot in our simulation of the EUCFLUX site. This was a slightly higher proportion than for





GPP, i.e. GPP was more impacted than transpiration by K deficit. As a consequence, the simulated WUE for GPP was higher

in +K condition in our simulations (Tab.4).

## 3.4   Sensitivity analysis



**Figure 7.** Sensitivity of GPP cumulated over a rotation to a ±10% change in parameters related to soil availability, diffusion resistances and response of leaves development to K. For each parameter, the sensitivity analysis was performed for the two contrasting K supply regimes (+K and oK). Note differences on the y-axes scales, for sake of clarity.





Sensitivity analysis was done separately for a K fertilised and the K omission stand. The simulated GPP cumulated over the whole simulation period of five years was highly sensitive to few sub-models parameters, but this sensitivity was strongly dependent on the fertilisation treatment (Fig.7a). Among the tested parameters, in the +K condition, GPP was sensitive to parameters related to the leaf phenology, especially the ones driving maximum leaf area and maximum leaf lifespan. Increase in maximum leaf area ($LA_{max}$, number of leaves produced by height increment ($\kappa$) and maximum lifespan ($LLS$) parameters resulted in GPP increases in the +K simulations. These parameters had an impact on the leaf area of trees, thus directly affecting photosynthetic area. This shows that under non-limiting K availability (+K conditions), the simulated GPP was mainly limited by leaves developmental aspects among the parameters tested here, i.e. the ones directly involved in processes related to K cycling.

Under severe K deficiency (oK), simulated GPP was sensitive to a greater number of parameters but a pattern still emerged. In this case, variations in the parameters controlling the values of K inputs to the ecosystem ($K_{atmosphere \rightarrow soil}$, $K_{litter}^{ini}$, $K_{soil}^{ini}$) produced a strong response in simulated GPP, highlighting the strong limitation of GPP by K availability. The amplitude of the response was in line with their respective contribution to the total amount of K entering into the system in the system throughout the rotation (Tab.2). For instance, a small increase in atmospheric deposition is accumulated through the entire rotation and has a larger impact that small changes in the initial value of K content in the litter or in the soil. In the oK condition, contrary to +K, the model was not sensitive to the parameter controlling maximum leaf lifespan ($LLS$, Fig.7. Indeed, the maximum leaf lifespan was almost never reached because of earlier leaf fall due to K limitation (Fig.4c). Other parameters ($t50_{LA}$, $k_{LA}$) controlling maximum leaf growth had also a much lower impact for similar reasons. Sensitivity of simulated GPP to the leaf maximum individual area ($LA_{max}$) in the oK stand was high, as in the +K case. Indeed, this parameter was used both in the +K and oK case because the area increment, depending on this target value, was modulated when a leaf cohort experienced a K deficit (eq. 2 and 29). This led to a variation in leaf area of each cohort which affected directly the GPP. The second most important leaf parameter in the oK stand was the resistance to K flux from the leaf to the phloem ($R_{leaf \rightarrow phloem}$, Fig.7b). This parameter was important since it controlled the competition for the K resource between new leaves (demanding K) and old leaves (providing K through resorption). In our simulations, an increase (+10% in Fig.7b) in resistance to K flux between the leaves and the phloem had a positive impact on GPP, at least in the range of values considered. Indeed, increasing the resistance ($R_{leaf \rightarrow phloem}$) led to a higher conservation of K in the leaves, which kept the leaf K concentration longer above the leaf shedding K threshold, which increased the leaf realised lifespan, which in turn increased canopy area. Since LAI in the oK stand was low (Fig.6a), a small increase in LAI can have an important impact on stand GPP. $[K]_{min}$ is the K concentration value below which leaves start their senescence. An increase of this value caused earlier leaf fall because this value was reached sooner, and GPP therefore decreased. Finally, the parameter related to symptoms area $SP_{max}$ was also sensible in the model, i.e. the GPP is reduced when the symptoms of area increase.





## 4 Discussion

In this work, we developed a process-based model simulating the influence of K on the gross primary productivity and transpiration fluxes of tropical eucalypt plantations. Such models have rarely been published in the literature, and we identified it "a worthwhile endeavour" (Reed et al., 2015) owing to the importance of K limitation of productivity in forests around the world (Sardans and Peñuelas, 2015). We used tropical Eucalypt plantations as our primary study system, since nutrient limitation has been extensively studied there, they are typically highly fertilised, and K-omission experiments show a very strong response of

wood productivity to K deficiency (Laclau et al., 2010). Our K model incorporates parts of the K cycle that were essential in determining K availability at the plant level. We focused on the modelling of the carbon-source activity on canopy processes and fluxes, starting with the demography of leaves and the impact of K availability on their functioning. In particular, we first considered the impacts of K on leaf development, photosynthetic capacity and senescence. We included processes that we identified as central (Cornut et al., 2021) regarding the K-limitation of GPP in these plantations. While adding processes to a

mechanistic model is pertinent from a realism perspective, one must consider if the implementation of new processes increases or decreases the predictive power of the model in a given context (Famiglietti et al., 2021). Here, the model additions were clearly necessary since the CASTANEA model, into which we developed the K modules, was initially incapable of reproducing the effect of K limitation on GPP and no mechanistic model of the effect of K on plant productivity at the stand level existed. This development also broadly followed several of the guidelines posited by Famiglietti et al. (2021) in their paper address-

ing the question of models' structural complexity: 1) the use of datasets (here multiple experiments over multiple rotations) to constrain model parameters, 2) the new developments led to increased forecast ability (since no forecast of K deficiency was previously possible), and 3) we sought to calibrate unmeasured parameters. We adopted a reductionist approach, typical of the development of mechanistic model, by formulating and parameterising the model on dedicated experiments conducted at the organ scale. Only a few parameters were calibrated on carbon and water fluxes measured at the ecosystem scale. It is

noticeable that the model was parameterized in a fully fertilised stand, and then allowed to run in a virtual K omission stand with, as the only difference, a reduced amount of K fertiliser brought the first months after planting. The simulations showed a strong impact on C and water fluxes.

For K supplied trees, our model was able to simulate GPP and water fluxes close to the measured flux values at the EU-CFLUX experimental site, both in terms of seasonality and magnitude (fig.6). Compared to our model, the MAESPA model

uses a much finer spatial scale to model water and carbon fluxes (Christina et al., 2018, 2015) in both fertilisation regimes, but the parameterisation of the model is different in +K and oK, i.e. without simulating the K cycling and its impact on the parameters (which is a feature of CASTANEA-MAESPA-K). It is possible that the CASTANEA-MAESPA-K model presented here lost some accuracy in the prediction of carbon and water fluxes compared to MAESPA alone, due to the inclusion of new processes linked to K cycle instead of a direct parameter forcing with measurements. It also did not use the 3D representation

of trees of MAESPA which had probably improved the simulation of fluxes during the first year of the rotation, before canopy closure (Christina et al., 2018). At the rotation scale, however, the differences between the measured accumulated GPP and the simulated accumulated GPP flux are small (Fig.S1).



The difference in cumulated GPP between the +K and oK stands simulated by the model was large on average, but varied during the rotation. In the first year the difference in GPP between oK and +K (table 3) was underestimated in our model compared to Christina et al. (2015). The leaf cohort model also showed that leaves were not K-limited at the beginning of the oK stand rotation (Fig.2c). Both leaf K content and symptomatic leaf area showed similar dynamics between the simulated oK and +K stands until around 1 year of age (Fig.6b). These results suggest that until this time, K was not more limiting in oK than in +K. The simulated plant available K in the soil was similar in both treatments at the beginning of the rotations, which suggests that either K availability was in fact high at the beginning of the oK simulation (through litter remaining at harvest and K available in the soil from the previous rotation) and/or that our model overestimated K soil access the first year.

The simulated water-use efficiencies (WUE, Tab.4) were in the range of the spectrum for C3 woody plants (Lambers and Oliveira, 2019) and resulted from simulated transpiration and GPP fluxes that compared well with observations (Tab.4 and 3). Our simulations showed a decrease of both GPP and transpiration in the oK stand that was consistent with evidence from the MAESPA model (Christina et al., 2015, 2018) and from experiments (Epron et al., 2012). WUE was higher in the simulated +K treatment than in the oK treatment (Tab.4). This result from our model simulations was in accordance with experimental data on wood WUE but differed from experimental data on leaf intrinsic WUE, for which no effect was reported (Battie-Laclau et al., 2016). The observed difference was was small and in their study, Battie-Laclau et al. (2016) explained the difference between wood WUE and intrinsic leaf WUE by differences in the post-GPP processes of carbon allocation in +K vs. oK. This could suggest that our approach of restricting the effects of symptoms to the photosynthetic capacity was insufficient and that a direct effect on stomatal response or mesophyll conductance would be necessary for our model to agree with experimental data. Part of the effects of K on leaf functioning could also be ignored by our approach of direct proportionality between the area of symptoms and the reduction of leaf photosynthetic capacity. Studying the response of leaf functioning to a gradient of individual leaf K content (Basile et al., 2003; Shen et al., 2018) may be useful to diminish the uncertainty regarding this response (see section 6.6).

The submodel we implemented for the simulation of plant K uptake was a simple demand model, dampened by a resistance meant to represent diffusion and sorption/desorption processes that impede the uptake of K ions by the plany from the soil. It was similar to models used successfully in ForNBM (Zhu et al., 2003) and ForSVA (Arp and Oja, 1997), that are based on the law of diminishing returns (van den Driessche, 1974). Except for the soil access equation (eq. 10), our model did not consider K uptake kinetics to depend on root density. This was in part due to the highly dynamic growth of eucalypt trees, that go from saplings to 25-30m trees in less than six years (the same being true for roots down to 16-m depth (Christina et al., 2011)). However, the sensitivity analysis showed that GPP was not greatly affected by the resistance to uptake in both fertilisation conditions. This is in accordance with results from the Itatinga site, where $K^+$ ions appeared weakly sorbed to this sandy soil, hence the process of diffusion was probably not limiting (Cornut et al., 2021). This could be further amplified by the fact that uptake of K by roots can take place directly in the litter (Laclau et al., 2004) thus bypassing the soil entirely. Taking K-soil interactions into account might however be necessary if one were studying leaching of K ions in the soil.

The importance of the accurate measurement of K sources in the system was underlined by the results of the sensitivity analysis. The simulated GPP of the oK stand was sensitive to variables relating to K inputs. The GPP showed a strong response



to small changes in the yearly influx from atmospheric deposition. This mirrors modelling results that show a strong response
of NEP to increasing N deposition (Zaehle and Friend, 2010; Dezi et al., 2010). The response of GPP to initial K litter stock
underlined the importance of harvest residues in the maintenance of plot fertility. It was apparent that the model shifted from
developmental (leaf production, maximum leaf lifespan, leaf area) and pedoclimatic limitations of GPP in the +K treatment
to biogeochemical limitations in oK. For example, the level of mineral weathering had an important impact on the GPP flux
of the oK stand (not shown here, see Part 2), but uncertainty regarding this flux is high (Cornut et al., 2021; Pradier et al.,
2017; de Oliveira et al., 2021). We believe these results confirm the importance of studying biological weathering of minerals
in situations of strong K limitation in forests.

    The analysis of the model also showed the importance of internal K cycling, especially the resorption flux between the
leaves and the phloem. The intense cycling of K in plants has been amply demonstrated (Marschner and Cakmak, 1989).
Measurements are still lacking to evaluate whether our phloem demand simplification to explain the variation in leaf K content
is true in different conditions.

**5  Conclusions and perspectives**

This study is the first attempt to simulate the K cycle in a forest ecosystem, and its intimate link to the carbon and water balances
at different time and space scales. It was developed based on data and processes observed in eucalypt fast-growing plantations
under contrasting fertilisation regimes. The model was tested against stand-scale measurements and showed reliable results for
both K-fertilised and K-omission simulations. First analysis show that K amounts present at the beginning of the rotation (in
litter, soil or fertilisation) and atmospheric deposition are essential to explain the overall amounts of K in foliage. Then the
internal K cycling dominates the K availability to leaves, which in turn influence strongly leaf development, leaf area index
and GPP.

    The coupled Carbon-water-Potassium forest process-based model developed in this study represents an important step in
the endeavour to understand the nutrient limitation of forest productivity. This study, focusing on the canopy and C source
processes is followed by a second part (in a companion paper) which will investigate the C-sink limitation of growth under low
K availability. It also provides a framework for the development of modules that will incorporate other ionic nutrients such as
Mg or Ca. The leaf cohort model developed is also a good starting point for accurately simulating nutrient fluxes in tropical
forests that follow a continuous phenology. These modelling frameworks can then be adapted to other similar systems. This
work was enabled by long-term omission experiments and detailed data collection at these sites (Cornut et al., 2021). This
further underlines the necessity of these stand scale manipulation experiments for nutrient modelling work.

*Author contributions.* IC carried out the devloment of the model as well as wrote the original draft of the manuscript. GlM and ND supervised
the work, participated in the conceptualization of the model and reviewed the original draft of the manuscript. JPL, YN, JG participated in





the acquisiton of the data and reviewed the original draft of the manuscript. VFD carried out the photosynthesis experiments. All authors provided critical feedback and helped shape the research, analysis and manuscript.

*Competing interests.* The contact author has declared that none of the authors has any competing interests.

*Data availability.* Data is not freely available due to private funding of experimental sites but is available upon request.

*Acknowledgements.* Ivan Cornut was funded by the ANR under the "Investissements d'avenir" programme with the reference ANR-16-CONV-0003 (CLAND) and by the Centre de coopération Internationale en Recherche Agronomique pour le Développement (CIRAD). The data acquired on Eucalyptus stands at Itatinga station, Brazil, and partly re-analysed here, were funded by Universidade de São Paulo,
CIRAD, Agence Nationale de la Recherche (MACACC project ANR-13-AGRO-0005, Viabilité et Adaptation des Ecosystèmes Productifs, Territoires et Ressources face aux Changements Globaux AGROBIOSPHERE 2013 program), Agropolis Foundation (program "Investisse-ments d'avenir " ANR-10-LabX-0001-01) and from the support of the Brazilian state (Programa de Cooperacão internacional capes/Fun-dacão AGROPOLIS 017/2013'). We are grateful to the staff at the Itatinga Experimental Station, in particular Rildo Moreira e Moreira (Esalq, USP) and Eder Araujo da Silva (http://www.floragroapoio.com.br) for their technical support. EUCFLUX 1 project was a coopera-
tive program with participation of Arcelor Mittal, Cenibra, Bahia Specialty Cellulose, Duratex, Fibria, International Paper, Klabin, Suzano, and Vallourec Florestal, coordinated by the Forestry Science and research Institute - IPEF (https://www.ipef.br/). The data acquired on the response of photosynthesis to leaf K were funded and conducted by Suzano.





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
