# Peer review of "Potassium-limitation of forest productivity, part 1: A mechanistic model simulating the effects of potassium availability on canopy carbon and water fluxes in tropical eucalypt stands"

_EGUsphere, 2022_

## Author Response (AR1)

Cornut et al present a description paper of a process-oriented model of an Eucalypt plantation with the major novelty of accounting for potassium cycling in an explicit way. The model is calibrated and evaluated based on data from a fertiliser trial in Brazil, and model predictions for potassium fluxes are described.

This is a timely and important endeavour and presents a challenging exercise. While the work is important and could provide an important step forward, there is a lack of attention paid to the description of the model calibration, separation of model evaluation from pure predictions, and the writing. Besides, there are some questions about the appropriateness of model assumptions.

*We thank Reviewer#1 for her/his thorough review and detailed comments of our article. In the following we address the main comments that refer to hypotheses, theory or interpretation of results.*

**Major points:**

**Appropriate of model assumption:**

It is surprising that potassium leaching is observed to be negligible (L296) while potassium is assumed to be a highly water transported element in your model. How can there be no leaching of the K+ experiment if potassium is added such that plants are non K limited?

Doesn't the modelled accumulation of soil K during the experiment (Fig 4) suggest the assumption is invalid? Are there site observations available which could indicate such an accumulation is realistic?

*The simulated accumulation of K in the soil during the experiment is consistent with the high levels of fertilization at both the Itatinga and Eucflux sites. These fertilization levels are slightly above the necessary levels for optimal plant growth since they were chosen to make sure that K is non-limiting. The absence of deep soil leaching of K at our site is indeed counter-intuitive since it is a mobile nutrient, the soil is sandy, and that large amounts of K were applied. However, the very deep soils combined with the capacity of the soil CEC to retain K+ ions (table 22 in Maquere, 2008 ,see below Fig.1), and the uptake and storage in tree biomass led to an absence of measurable leaching fluxes below a depth of 3m.  Ceramic cups set up down to a depth of 3 m and soil solution sampling continuously over the first three years after planting in this experiment (unpublished data) have shown that K+ fluxes were < 5 kg ha-1 yr-1. This is the case both at the Itatinga site (Maquere, 2008) and at the Eucflux site  (Caldeira et al., 2023, FE&M, in press). Even after the clear cutting of the plantation, no K leaching fluxes below 3m were measured at the Eucflux site (Caldeira et al., 2023) and at Itatinga.*

**Table 22** Estimation of the capacity of the soil (in kg ha$^{-1}$) to retain cation or anion by non specific adsorption (CEC and AEC) or specific adsorption. For each element, the capacity is calculated for each soil layer from the bulk density of Table 11 and the CEC or AEC of Table 20 fully saturated with the studied element. The charge attributed to each element is +1 for Na, K and N-NH$_4$, +2 for Ca and Mg, +3 for Al, -1 for N-NO$_3$, Cl and P-H$_2$PO$_4$, and -2 for S-SO$_4$. Specific adsorptions are calculated from Table 21. n.d.=not determined.

| | \multicolumn CEC | | | | | | \multicolumn AEC | | | | \multicolumn Specific adsorption | |
| --- | --- | --- | --- | --- | --- | --- | --- | --- | --- | --- | --- | --- |
| | Na | K | Ca | Mg | Al | N-NH$_4$ | N-NO$_3$ | Cl | S-SO$_4$ | P-PO$_4$ | S-SO$_4$ | P-PO$_4$ |
| | | | | | | | kg ha$^{-1}$ | | | | | |
| 0-5 cm | 511 | 869 | 446 | 270 | 200 | 311 | 9 | 23 | 10 | 20 | 0 | 232 |
| 5-15 cm | 378 | 642 | 329 | 200 | 148 | 230 | 15 | 38 | 17 | 33 | n.d. | n.d. |
| 15-30 cm | 443 | 753 | 386 | 234 | 173 | 270 | 27 | 69 | 31 | 61 | n.d. | n.d. |
| 30-50 cm | 628 | 1067 | 547 | 332 | 246 | 382 | 53 | 134 | 61 | 117 | n.d. | n.d. |
| 50-100 cm | 1394 | 2371 | 1215 | 737 | 545 | 849 | 132 | 334 | 151 | 292 | n.d. | n.d. |
| 100-200 cm | 2270 | 3861 | 1979 | 1200 | 888 | 1382 | 578 | 1464 | 662 | 1279 | n.d. | n.d. |
| 200-300 cm | 1852 | 3150 | 1614 | 979 | 725 | 1128 | 1031 | 2611 | 118 | 228 | 2996 | 6405 |

*Figure 1: Estimation of cation and anion retention capacity of the soil at the Itatinga site. This table was reproduced from Maquere (2008).*

I could imagine that potassium might be efficiently adsorbed to organic matter preventing leaching losses? But if this is the case, why is it omitted in the model? If so, you should explain why this was omitted, and what the implications for the result are.

*This is the case at our site since a large part of the cationic exchange capacity was due to the organic matter in the soil in the upper layers (Maquere, 2008). This was omitted in the model since no deep leaching fluxes of K were measured even at high (higher than practiced in commercial plantations) levels of fertilization (Caldeira et al., 2023), and therefore this mechanism could not be calibrated. Furthermore, the model did not consider this level of details as concerns K exchanges between the soil and the soil solution. In the future, an improvement in model genericity would require a model of K flux and exchange in the soil since other sites could present deep leaching loss of K. This has very few implications for our sites but could lead to unrealistic simulated accumulation of K in the soil at sites with shallower soils or soils with less cationic exchange capacity. This is discussed on lines 755-760 of the revised manuscript.*

**Model description:** Not all fluxes are described with equations (e.g. Kleaf→litter Is missing ) and not all changes in K pools are described (e.g. Ksoil or K in roots ). Make sure all fluxes and pools are described. The overview figure is very hard to follow (see minor points below). The coupling of the water cycles is not described (see minor points).

*We thank Reviewer#1 for pointing out these inconsistencies and we address these points below in our answers to the "minor points". Kleaf to litter is now described in equation 27 at line 463. K pools are not described here but are described in the companion paper (Cornut et al., 2023), a reference to the companion paper was added on line 276.*

**Description of model calibration:** There is hardly any information on how the calibration of parameters was achieved. e.g. what method was used, what data was used for a given parameter. Where does the data origin from, etc. It is not clear if Fig2 shows the results of model calibration or an evaluation (as suggested on Line 555).

*Most of the processes were parameterized based on dedicated experiments, as described throughout the text and in tables S1, S2 and S3. References to these tables were added when necessary in the revised manuscript. When calibration was necessary, it was done at the process level and not at stand level. For example, the parameters of the leaf expansion model were fitted on leaf expansion data measured on this site. However, we agree that some descriptions were lacking and we have changed the text to detail explicitly how the parameters were obtained, for each process represented in the model. When calibration of parameters was necessary, it was achieved using a linear exploration of the parameter space and evaluating model fit using RMSE. Figure 2 was used mainly to show the theoretical functioning of the leaf expansion model without the rest of the model. In the current version of our manuscript, this figure was modified to show two simulated leaf cohort during respectively a +K plot and oK plot simulation.*

**Lack of model evaluation.** The results are mostly describing model results with little confrontation with observation, etc. There are comparisons of model predictions and observations but they fail to identify and highlight predictions which are apparent results of the model and which are calibrated. The discussion would benefit from the restructuring into distinct parts for evaluation and for prediction. Besides, all datasets and their purpose (evaluation, calibration) should be described in the method section, e.g. only in the discussion the Christina et al 2015 model data is explained.

*In this manuscript, which consists of part 1 of a two-paper article series, we focused on parameterising the model using data from both study sites. This was done since the data is incomplete at each site. Carbon and water flux data were only acquired at the Eucflux site while the growth response of eucalypts to K omission was only measured at the Itatinga site. Calibrations were done at the scale of processes and not for the whole stand. For example, leaf production in the fully fertilised condition was calibrated by using LAI, biomass and litterfall data.*

*We added the following paragraph at lines 550-554: "The calibration of the simulated processes was only done in the +K condition since the responses of different processes to K deficiency were derived from measured parameters (except for the leaf expansion process which was calibrated in both +K and oK conditions using Battie-Laclau et al., 2013). This meant that oK simulations were meant to act as tests for the model as a whole by seeing how the model was able to replicate the response of the canopy or fluxes to K deficiency".*

**minor**

Section 2.3

This section is mostly focused on the motivation of revising the water cycle in CASTANEA than in describing what has been actually done, i.e. the new model structure of CASTANEA-MAESPA. It is not clear how the coupling has been achieved. I would suggest explicitly stating the modifications done to the underlying equation of CASTANEA given the scope of the paper as a model description and reference paper.

*Thank you for this suggestion. The coupling was made by integrating MAESPA sub-routines in the CASTANEA code. The sub-routines were those related to soil water and photosynthesis. Radiation and water interception were simulated by CASTANEA; and the integrated sub routines from MAESPA simulated photosynthesis, transpiration and leaf water potential for each canopy layer. Soil water fluxes and water potential were calculated using sub-routines from MAESPA. A general model overview schematic was added in the supplementary material as Fig.S1 and the following was added on line 162 of the revised manuscript: "The modules of CASTANEA simulating light interception, water interception, carbon allocation and the growth of organs and organ respiration were coupled with the modules of MAESPA simulating soil water dynamics, leaf photosynthesis, transpiration, and plant hydraulics (Fig.S1). More precisely, the coupled version includes:*

*1. CASTANEA computes the diffuse and direct incoming radiation reaching sun and shade leaves of a canopy layer*
*2. This radiation is used in MAESPA to compute leaf-scale carbon and water processes, based on what is done usually at voxel-scale in MAESPA*
*3. Net photosynthesis is calculated by MAESPA per canopy layer and summed up at the scale of the canopy,*
*4. then CASTANEA simulates the carbon allocation to the different organs, the organ respiration and their resulting growth,*
*5. CASTANEA computes the rainfall interception and throughfall, and therefore the water entering in the soil, and MAESPA continues the water cycle simulation with water infiltration in the soil, evaporation, water uptake from different soil layers and water table, transpiration, water potential in the soil, roots and leaves, and impact of leaf water potential on stomatal conductance"*

You should indicate units of all variables. Use a consistent format for units, e.g. there is am ic of /year and year-1

*Thank you for your attentive review, -we checked all units in the manuscript and homogeneized notations throughout the manuscript.*

Figure 1: An overview figure is an excellent idea but the current figure is hard to follow.

- What does the broken line stand for? What do the different colours stand for?

- Caption indicates all K fluxes are based on Ohm's law which isn't the case. Rephrase.

- The figure is a mix of process, fluxes, relationships, pools. E.g .you could produce separate figures/panel: one for pool & fluxes, and one for the process linkage

*The figure has been modified accordingly and is now split into two panels: 1) canopy processes 2) K fluxes and pools. The caption was also adapted to explain the difference between black, purple and dashed arrows.*

Line 5: large-scale - specify what 'large' means here

*The text was modified and reads "at the stand scale" instead (L.5)*

Line 10: internal/external is not clear unless you define the boundary of your system. I would suggest to rephrase

*We have rephrased as inputs to the stands and internal cycling. (L. 11)*

Line 68: It is not clear why it is a prerequisite one could also theoretically start modelling with the sinks than with the source.

*This is indeed possible, but modelling C-sources is well documented thanks to the good theoretical framework surrounding photosynthesis (Farquhar model) and stomatal response (Ball and Berry model, Tuzet model, etc.). Driving the C-source activity and stomatal functioning by C-sinks has been attempted with some success (Hölttä et al., 2018) but is more computationally complex and had never been calibrated on eucalypts. We have added these arguments to the new version of the manuscript. To insist on this aspect, the lines 68-73 were modified to: "The combined influences of K on C-source and C-sink processes explain the K limitation of productivity. The present study focuses on modelling the influence of K on the C-source (i.e. on GPP), which is based on the assumption that C-source modelling would be the most straightforward step to start modelling the K limitation on productivity. Indeed, process-based models of the C-source activity have been developed for more than four decades (Farquhar et al., 1980), which contrasts with models representing the activity of C-sinks (e.g. (Hölttä et al., 2006)) which, while relevant (e.g. Guillemot et al., 2017; Körner, 2015), are relatively new and have not been validated at a large scale."*

Line 108: specify to what extent this clone is comparable to the other one?

*Most of the clones planted in this region are very similar, because they were all selected locally for the same climate and main soil properties. For instance, the wood production is similar, leaf area index evolves in the same ranges of values, photosynthetic parameters are similar (unpublished data). However, they also differ for some other aspects such as branches and litter turnover, stomatal conductance, etc. However, differences between clones can be hard to investigate at our sites since they were not planted at the same time and thus did not experience the exact same climatic/edaphic conditions at the same developmental stage. This was addressed at line 121 of the revised manuscript.*

Line 126: is this a novel technique ? Give references or additional information on how you derived the damaged leaf area.

*The technique was developed in the frame of the present study. It is based on simple color threshold on leaf scans. Indeed, symptoms areas are clearly different in colours in the visible range and observable by photointerpretation. Color thresholds were therefore adjusted manually. We will add some more description: "… based on a colour threshold calibrated by photointerpretation and automatized in a Matlab ® script". A reference to a dataverse repository containing example data and algorithms was added at lines 131-132 of the revised manuscript.*

L167: does this mean you have (365 days *6 years ->) 21190 leaf cohorts at the end of a 6 year rotation ? Is this really needed?

*This would not be needed to simulate leaf dynamics but is useful for the simulation of K fluxes between leaves and the other tree components. Once all leaves of the cohort have fallen the cohort is no longer simulated. So there are no more than 400 cohorts (since leaves have a 400 day theoretical lifespan) at the same time. While this seems a lot, this is in fact easier to simulate (at the expense of some memory space – but nothing critical) than grouping leaf emergence and growth every x days experiencing various weather and soil conditions. Simulating daily cohorts also brings stability to the model since all processes are simulated at a daily scale (carbon and K fluxes). The added computation brought by these cohorts is also negligible compared to the half-hourly*

*calculation of photosynthesis and transpiration for each canopy layer (since it results from a computationally intensive minimum search).*

L185: m2 of ground ? leaf?

*Of ground, thank you.*

L 187: what is P_leaf ? ; units of k are missing

*We corrected this error. P_leaf now reads as N and the units of K have been added on line 199 of the revised manuscript.*

L188: indicate how the calibration was performed (which obs variable did you target, time step, method of calibration, etc)

*The calibration was a linear exploration of parameter space using multiple normalised RMSE as a goodness-of-fit indicator. The data used for calibration were destructive leaf biomasses, leaf area, leaf biomass and leaf fall measurements. We used mainly cumulative leaf production and leaf-fall as the points to fit since simulating fine weekly variation in leaf production or leaf fall were not the objectives here. The time-step between these measurements were different (yearly, monthly). Description of the calibration process was added at line 201 of the revised manuscript. Multiple normalised RMSE was defined as eq.S1 in Supplementary material of the revised manuscript.*

L190cc: equation/description for leaf fall is missing

*The following sentence was added at line 186 of the revised manuscript: Leaf fall occurred when leaves reached a fixed K minimum threshold or the end of their lifespan.*

Section 2.4.4. : explain and show how this equation was fitted.

*This equation was fitted using leaf expansion measurements in both fully fertilised and K omission stands (see Battie-Laclau et al., 2013). These measures were conducted on 70 tagged leaves from creation to full-expansion. These details were added at line 225 of the revised manuscript.*

L245 : indicate how is alpha computed. Is it a fixed input parameter?

*Yes alpha is a fixed input parameter and was calibrated using fine scaled leaf K concentration measurements (Laclau et al., 2009). More generally the source of the parameters is described in table S3.*

L253: which cycle? You mean ecosystem?

*"Ecosystem cycle" is better, thank you for the suggestion.*

L264: is there no biological mediated K release from litter?

*We found no evidence of biologically mediated K release from litter in the literature and the difference between the dynamics of K and N or P (which are known to be biologically mediated) in the litter suggest that this is not the case at our sites. Furthermore, K losses are similar (Fig. 2) in leaves and branches contrary to N or P (Laclau et al., 2010; Maquere, 2008) which suggests that litter leaching is the most parsimonious explanation for the dynamics of K in the litter. The following paragraph was added at line 294 of the revised manuscript: there was no simulated biologically mediated K release from the litter since no reference to this process was found in the*

*literature. Moreover, measures of K concentration in the litter of leaves, branches and bark all decreased exponentially at the same rate (Maquere, 2008). This was not the case for N and P (known for their biologically mediated release), indicating that K release from the litter is indeed mainly the result of leaching.*

[Figure]

*Figure 2: Litter decomposition at our study site. Decomposition dynamics of dry matter (A), Nitrogen (B), Phosphorus (C) and Potassium (D) are shown here. The x-axis is time (months) since the litterbag was put on the ground. The y-axis is the amount of remaining DM, N, P or K in %. Folhas, Galhos finos and Galhos grossos are respectively leaves, fine branches and large branches. This figure was reproduced from Maquère (2008).*

What about the unavailable soil K. indicate how this was represented.

*Unavailable soil K is the pool of K not reached by the root system, especially at the beginning of the rotation when planted trees are small. Unavailable soil K was represented as a pool that progressively was added to the K accessible soil K using equations of horizontal root expansion (equation on line 332). Inputs of K reaching the soil were shared between available soil K and unavailable soil K depending on their respective relative surfaces.*

What about root and wood litter production? Was this omitted?

*Wood litter production was omitted since the model system are young eucalypt plantations with no mortality and trunk wood is exported at harvest. Branches mortality and bark litter were however simulated. Root litter is simulated, but not described in this Part 1 but is described in the companion paper (Cornut et al. 2023) and was modelled in the simulations shown here. For root litter and branch litter we used measured turnover rates (using minirhizotrons for fine roots and biomass and litterfall data for branches). The following sentence was added for clarification at line 284 of the revised manuscript: "The flux of K from branch, bark and fine root to litter was simulated but is not described here (see Cornut et al., 2023)."*

Is K immobilisation by soil organisms really negligible? The initial loss from litter might be due to leaching, but the question is rather how much of all the K in litter is lost via leaching. Can you elaborate on this.

*When looking at the K dynamics in litter (see Fig.2 from Maquere, 2008) it is probable that biologically driven decomposition processes are only responsible for a small fraction of the K losses. This is visible when comparing K losses to N and P losses (since N and P follow the same dynamic as dry matter). We didn't find any information regarding immobilisation of K by soil organisms. This could be the result of the negligible effect of soil micro-organisms on the cycle of K in the soil or a measurement/publication bias.*

L326: why not call it maximum K conc instead of optimal K? Can you rule out that the optimal conc < max conc?

*The difference is that there could be luxury consumption or storage of K in the leaves, therefore maximal K concentration is not necessary the optimal one. There is a difference (Walker et al., 1996) between K stored in vacuoles (very variable) and cytosol (less variable) but we cannot conclude that the variability of K in the vacuole is evidence of luxury consumption or otherwise.*

L354: which 'part 2'?

*This manuscript has a companion paper we called "Part 2", the full reference is: Cornut, I., le Maire, G., Laclau, J. P., Guillemot, J., Nouvellon, Y., & Delpierre, N. (2023). Potassium-limitation of forest productivity, part 2: CASTANEA-MAESPA-K shows a reduction in photosynthesis rather than a stoichiometric limitation of tissue formation. EGUsphere, 1-27. In the revised manuscript every reference to part 2 was changed to read Cornut et al., 2023 in a clarification attempt.*

L429: what is the significance of the speed of senescences for the equation?

*The speed of resorption, is a measure of how fast the K in the leaf can be remobilized to the phloem at leaf senescence. This process is very fast and it is possible that K is necessary for the remobilisation of sugars from the senescing leaves.*

L452: explain how K affects the wood production in this paper.

*A sentence briefly explaining how K can affect wood production was added at line 480 of the revised manuscript: "Briefly, tree trunk production could be affected by a reduced allocation of C*

*due to either a decrease in GPP or an increase in the share of C partitioned to other organs (for more details on trunk production see Cornut et al., 2023)."*

L453: impact on what?

*Impact on the generation of new leaves, this was added on line 485 of the revised manuscript: "on the number of new leaves generated"*

L472: you mean 'was replaced with'?

*We meant that leaf expansion was recalculated using an updated value for the expansion. The line (504 of the revised manuscript) was modified to: Secondly, leaf water content Wleaf was recalculated using an updated value for the expansion ($W_{xylem \to leaf}$ instead of $W_{xylem \to leaf}$).*

L535: indicate over which period. Does this refer to Table 2?

*We rephrased this to: "Ecosystem K cycle during a rotation" (L. 587)*

L549: important for what ? you mean higher?

*Yes, "higher". (L. 604)*

L580-600: does the good agreement with Christina et al 2015 mean we don't need a potassium model to capture GPP and transpiration? The motivation for comparing your results with the ones of Christina et al 2015 should be given in the methods. Also a description of the data from Christina et al 2015.

*We compared our results to the results of the model in Christina et al. 2015 since the potassium effect that they simulate is not the result of a mechanistic modelling approach (which we use in CASTANEA-MAESPA-K) but two distinct parametrisation sets (one set for +K and another parameter set for oK). Our model only changes for the initial K fertilization amount parameter, all the processes included in the model now simulate the differences between treatments. The advantage of our model is the increased genericity, the feedback between K availability and growth, the capacity to simulate a fertilisation gradient (e.g. see the companion paper) and a decrease in computation time. These arguments were added in the section 2.9 Model Intercomparison of the revised manuscript.*

L598: why was it done for both? The K+ treatment effectively shuts off most of the model developments and is thus not really informative. It makes sense to report for traceability of impact of model developments, but might be better off in the SI as this is mostly relevant for MEASPE developers.

*Indeed, we found it useful to show that the model was able to accurately replicate fluxes and behaviour of the eucalypt plantation with classical optimal fertilization. This seems logical, but many processes were added and need to be tested. However, this is clearly not enough, and it is the changes in the ecosystem after removing K fertilisation that is the targeted important validation. This approach is now described at line 550 of the revised manuscript.*

L674-678: WUE: you never defined the modelled WUE. Avoid comparing apples with oranges. (e.g. https://hal.archives-ouvertes.fr/hal-01606915)

*Thank you for this comment. Indeed the simulated WUE we mention in this paragraph is WUE_GPP (GPP/Transpiration). We agree that comparing this WUE to other WUE (intrinsic or wood) is not of the highest relevance However, we do not have any direct measures for WUE_GPP and we wished to highlight the responses of different WUEs to K deficiency. This was detailed and clarified in the revised manuscript at line 731.*

L678-679: K and GPP vs N and NEP - what is the connection?

*Sorry, we do not understand this question.*

**References:**

Caldeira , A., Krushe, A. V., Mareschal, L., da Silva, P., Nouvellon, Y., Campoe, O., ... & Laclau, J. P. Very Low Nutrient Losses by Deep Leaching after Clearcutting Commercial Eucalyptus Plantations in Brazil. (Preprint) *Available at SSRN 4270148*.

Cornut, I., le Maire, G., Laclau, J. P., Guillemot, J., Nouvellon, Y., & Delpierre, N. (2023). Potassium-limitation of forest productivity, part 2: CASTANEA-MAESPA-K shows a reduction in photosynthesis rather than a stoichiometric limitation of tissue formation. *EGUsphere*, 1-27.

Hölttä, T., Lintunen, A., Chan, T., Mäkelä, A., & Nikinmaa, E. (2017). A steady-state stomatal model of balanced leaf gas exchange, hydraulics and maximal source–sink flux. *Tree physiology*, *37*(7), 851-868.

Laclau, J. P., Ranger, J., de Moraes Gonçalves, J. L., Maquère, V., Krusche, A. V., M'Bou, A. T., ... & Deleporte, P. (2010). Biogeochemical cycles of nutrients in tropical Eucalyptus plantations: main features shown by intensive monitoring in Congo and Brazil. *Forest ecology and management*, *259*(9), 1771-1785.

Laclau, J. P., Almeida, J. C., Goncalves, J. L. M., Saint-Andre, L., Ventura, M., Ranger, J., ... & Nouvellon, Y. (2009). Influence of nitrogen and potassium fertilization on leaf lifespan and allocation of above-ground growth in Eucalyptus plantations. *Tree physiology*, *29*(1), 111-124.

Maquere, V. (2008). *Dynamics of mineral elements under a fast-growing eucalyptus plantation in Brazil. Implications for soil sustainability* (Doctoral dissertation, AgroParisTech).

Walker, D. J., Leigh, R. A., & Miller, A. J. (1996). Potassium homeostasis in vacuolate plant cells. *Proceedings of the National Academy of Sciences*, *93*(19), 10510-10514.

This work by Cornut et al. developed a K biogeochemical model based on the relative benefits of two processed-based models (i.e. MAESPA and CASTANEA). A lot of work has went into this model development, and the authors splitted the work into two manuscripts, with the current draft focusing on carbon and water fluxes simulations, and the second draft focusing on growth limitation. I appreciate the reason to do so. In this review, I provide my comments specifically to the first part of their work.

*We thank Reviewer#2 for his/her review and for approving our choice of splitting the work in two manuscripts.*

In this manuscript, the authors described the mathematical formulations of the K cycle, the coupling of MAESPA and CASTANEA, and model parameterization and evaluation, including some sensitivity tests. Here, MAESPA served as the canopy model and CASTANEA served as the ecosystem C model. The rationale as to why to integrate the two models were well described (L143 – 155), but the details on how the two models were merged were quite lacking. For example, it's unclear how the 3-d structure of MAESPA was simplified into the 1-d structure of CASTANEA. It's unclear how leaf photosynthesis and transpiration of MAESPA was integrated with the light interception component of CASTANEA. Etc. Considering the vague information, I can't help but wonder if the authors actually ran both models but used the output of one to feed into the other. I suspect not, but I think the authors should further elaborate details on how the two models were merged.

*Since this remark came up in the comments of both reviewers #1 and #2 we have dedicated a paragraph to explaining the coupling between the two models. This is explained above in the answer to the comments of reviewer #1.*

Furthermore, abstract can be improved, as in many places the results are vague. For instance: "Simulations showed that K-deficiency limits GPP by more than 50% during a 6-year rotation, a value in agreement with the literature". What level of K-deficiency limits GPP by more than 50%, and what does the literature say in terms of uncertainty range? Is it the same species and stand? Moreover, "The negative effects of K-deficiency on canopy transpiration and water use efficiency were also reported and discussed". Can you be more specific and describe some key results and implications? Moreover, "Litter decomposition processes were of lower importance". This sparks readers interest to understand why, and I think it's useful to briefly describe your understanding regarding this "lower importance".

*Thank you for these suggestions for abstract improvement. We have found no information in the literature about the uncertainty range. The level of K deficiency that leads to this reduction is a total omission of K fertilizer in eucalypt stands. This is similar to measured GPP reduction at these stands (Epron et al., 2012). For the low importance of the litter leaching of K this is due to the very fast transfer of K from litter to the soil which means that this process does not immobilize a large quantity of K. The abstract was modified according to suggestions to include key results.*

Regarding the K cycle structure, I'm not sure how the mass balance for K was closed. The authors indicated that there are 7 pools of K, splitted into soil, soil fertilizer, litter, xylem, phloem, leaf and other plant organs. Can the authors describe how K was allocated in plants of different organ, and whether that matches with plant K uptake? In particular, I wonder why the authors did not consider allocation into root in their work? Did the authors consider the vertical growth of root and the associated K content at all? Furthermore, how soil K was mineralized and immobilized remains unclear. I suspect CASTANEA has a three soil organic matter pool structure for the soil component of the model, but this was not reflected in Figure 1. The process of plant litter entering soil and the associated biogeochemical processes should be better captured, or explained in the case of not included in this work.

*We thank you for your comments. We can assure that the mass balance of K is closed. This is not immediately visible in this manuscript but the allocation of K uptake to the different organs is described in detail in the companion paper (Cornut et al., 2023). Allocation into roots was considered and was based on objective functions similar to the ones used in the G'day model (Marsden et al., 2013). The process of K from the plant litter entering the soil was a leaching process (Cornut et al., 2022). This was chosen due to the mobile nature of K (that stands in opposition to N and P dynamics in soils) and what we believe is negligible interaction between K and decomposition processes (Maquere, 2008). This was clarified at line 300 of the revised manuscript. This was also one of the reasons why soil K dynamics were very coarsely described in CASTANEA-MAESPA-K. These choices were sufficient for our study but could prove a handicap for genericity. This is discussed at line 775 of the revised manuscript.*

Furthermore, this work introduces the limitation effect of K on many plant and ecosystem processes. Obviously, as the authors introduced, there are other limiting nutrients as well. In the current model structure, the authors did not consider the interactive effect of the relative limitation of N, P and K. I wonder if it is useful to discuss some of the potential influences on these interactive effects, and the challenges to actually implement them in a cohesive modelling framework?

*There is very little information pertaining to interactions between N or P and K. One could hypothesize a lower K demand when limited by N or P. N and P limitation might also lead to lower weathering in the rhizosphere (less enzymes being released, see N and phosphatases). We were also limited by the absence of a strong N or P limitation at our experimental sites. In the absence of these limitations modeling or testing for interaction between N, P and K is difficult.*

**Specific comments:**

L274" But there is a specific pool for bark, branch, so what K concentration did you assume for them?

*This is explained in the companion paper (Cornut et al., 2023) . Briefly, we assumed concentrations from destructive biomass and nutrient dosing measurements conducted at different ages during the rotation. A reference to the companion paper was added on line 274 of the revised manuscript.*

L276: What do you mean by "very lose K release rates"?

*Indeed this was unclear, this was reformulated as "very similar K loss rates" on line 312 of the revised manuscript*

L280: But K concentration in different plant organs are different, right? But in litter you assume a fixed concentration? How to close the concentration imbalance?

*We agree this was unclear in the manuscript, the concentration in litter directly depends on the concentration of the falling organs (after remobilisation for branches and leaves). Every day the litter pool is updated by adding the K mass of falling organs (computed as the actual K concentration of the falling organ multiplied by its biomass) and removing the losses that take place by K leaching from the littter. This was clarified in the manuscript by adding a reference to the companion paper for bark, branches and roots. The following sentence was also added at line 304 of the revised manuscript: Since we assumed the leaching rate was independent of the litter type, all simulated K litter was pooled into a unique K litter compartment (Klitter).*

**References**

Cornut, I., le Maire, G., Laclau, J. P., Guillemot, J., Nouvellon, Y., & Delpierre, N. (2023). Potassium-limitation of forest productivity, part 2: CASTANEA-MAESPA-K shows a reduction in photosynthesis rather than a stoichiometric limitation of tissue formation. *EGUsphere*, 1-27.

Epron, D., Laclau, J. P., Almeida, J. C., Gonçalves, J. L. M., Ponton, S., Sette Jr, C. R., ... & Nouvellon, Y. (2012). Do changes in carbon allocation account for the growth response to potassium and sodium applications in tropical Eucalyptus plantations?. *Tree physiology*, *32*(6), 667-679.

Marsden, C., Nouvellon, Y., Laclau, J. P., Corbeels, M., McMurtrie, R. E., Stape, J. L., ... & Le Maire, G. (2013). Modifying the G'DAY process-based model to simulate the spatial variability of Eucalyptus plantation growth on deep tropical soils. *Forest Ecology and Management*, *301*, 112-128.

Maquere, V. (2008). *Dynamics of mineral elements under a fast-growing eucalyptus plantation in Brazil. Implications for soil sustainability* (Doctoral dissertation, AgroParisTech), https://www.theses.fr/2008AGPT0086.